# A membrane-bound nuclease directly cleaves phage DNA during genome injection

Daniel S. Saxton[1], Peter C. DeWeirdt[1,2], Christopher R. Doering[1], Ian J. Roney[1] & Michael T. Laub[1,3 ✉]

From mammals to bacteria, the direct recognition and cleavage of viral nucleic acids is a potent defence strategy against viral infection, but it requires mechanisms for distinguishing self from non-self[1,2]. In bacteria, CRISPR–Cas and restriction-modification systems achieve this discrimination by recognizing specific DNA sequences or DNA modifications, respectively. Alternative mechanisms probably remain to be discovered. Here, we characterize SNIPE, an anti-bacteriophage defence system that constitutively localizes to the bacterial cell membrane in *Escherichia coli* to block phage λ infection. Using radiolabelled phage DNA and time-lapse microscopy to track phage genomes, we demonstrate that SNIPE directly cleaves phage DNA during genome injection. Based on proximity labelling, we find that SNIPE associates with host proteins essential for λ genome entry and with the λ tape measure protein, which facilitates λ genome injection across the inner membrane. SNIPE also defends against diverse siphoviruses, probably through direct interactions with their tape measure proteins. Our findings establish SNIPE as a widespread bacterial defence system that exploits the spatial organization of phage genome injection to specifically target viral DNA, representing a previously unknown strategy for distinguishing self from non-self in prokaryotic immune systems.

The ability to distinguish self from non-self is a fundamental feature of immune systems across all domains of life. In eukaryotes, pattern-recognition receptors enable this distinction by detecting conserved pathogen-associated features, such as lipopolysaccharides and flagellin[3–6]. Activated pattern-recognition receptors can trigger various immune responses, including the production of pro-inflammatory cytokines and the initiation of programmed cell death. Similarly, bacteria use abortive infection systems that recognize conserved features of invading bacteriophages and activate effectors that kill or arrest the host cell to prevent phage spread[7–10].

A second form of non-self recognition involves targeting foreign nucleic acids directly. In eukaryotes, the RNA interference pathway cleaves viral double-stranded RNA into small interfering RNAs that then guide Argonaute proteins to complementary RNA sequences, leading to the degradation of viral RNA[1,11]. In bacteria, various 'direct defence' systems also specifically target foreign DNA. This includes CRISPR–Cas systems and Argonautes that use guide RNAs or DNAs, respectively, to cut foreign DNA in a sequence-specific manner[1,2]. Furthermore, restriction-modification systems can recognize foreign DNA based on the presence or absence of DNA modifications[12]. Whether there are other mechanisms for directly identifying and degrading foreign nucleic acids has remained an open question.

## SNIPE is a membrane-bound nuclease

A previous genetic screen identified a direct defence system, provisionally named PD-λ-1, that potently blocks phage λ infection in *E. coli*[13].

For reasons described below, we renamed this system surface-associated nuclease inhibiting phage entry (SNIPE). To confirm that SNIPE provides direct defence, we infected cells at a concentration of phage λ such that approximately half of the cells were infected, and then monitored cell growth by time-lapse microscopy. In a population of cells lacking SNIPE, infected cells burst and caused a second round of successful phage infection in neighbouring cells, as expected (Fig. 1a and Supplementary Video 1). In sharp contrast to this, in a population of cells that did harbour SNIPE, no cell lysis was observed and cells grew to confluence. As a control, we also infected a population of cells containing an abortive infection system, PD-λ-3, and observed that infected cells lysed but prevented a second round of infection[13]. Moreover, growth-curve assays showed that SNIPE, but not PD-λ-3, enabled cell survival in the presence of high phage concentrations (Fig. 1b). These results confirm that SNIPE provides direct defence, enabling cells to ward off infection without compromising cell growth. Notably, SNIPE did not block plasmid DNA transformation, indicating that its activity is specific to phage infection (Extended Data Fig. 1a).

Analyses using AlphaFold and HHpred indicated that SNIPE is an elongated protein with a transmembrane domain near the amino terminus, a domain of unknown function (DUF4041) in the middle of the protein, and a GIY-YIG nuclease domain near the carboxy terminus (Fig. 1c and Extended Data Fig. 1b,c). A transmembrane topology model predicted that the N terminus of SNIPE is periplasmic, the single-pass transmembrane domain resides in the inner membrane, and the rest of the protein is cytoplasmic[14] (Extended Data Fig. 1d). We tested this model by inserting a PhoA–LacZα fusion protein at different locations

[1]Department of Biology, Massachusetts Institute of Technology, Cambridge, MA, USA. [2]Computational and Systems Biology Program, Massachusetts Institute of Technology, Cambridge, MA, USA. [3]Howard Hughes Medical Institute, Massachusetts Institute of Technology, Cambridge, MA, USA. ✉e-mail: laub@mit.edu

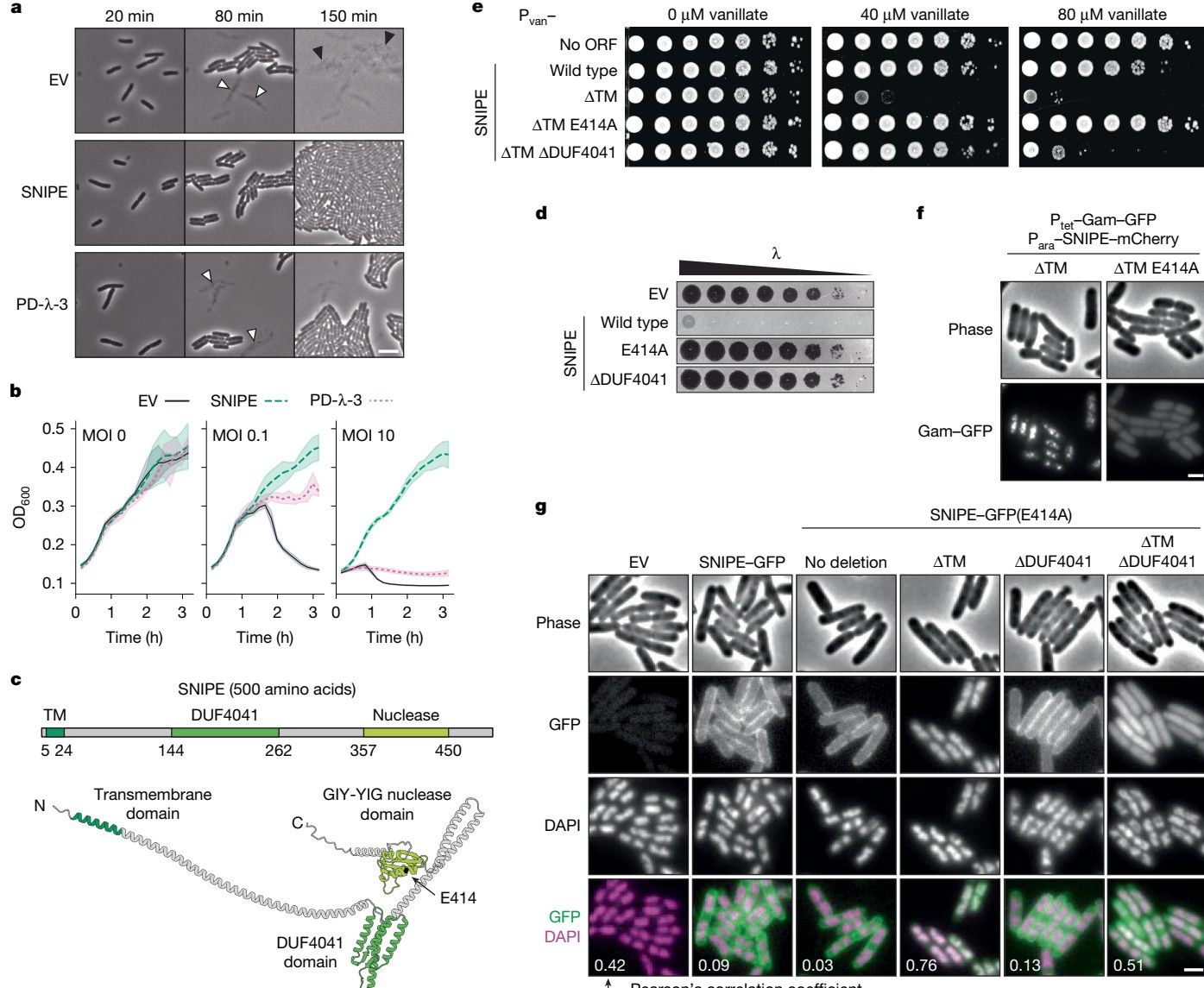

**Fig. 1 | SNIPE is a membrane-bound nuclease that provides direct defence against phage. a**, Time course of cells expressing an empty vector (EV), SNIPE or PD-λ-3. About half of the cells were infected with λ at time $t = 0$ min. White arrows show cell death from the initial round of infection and black arrows show cell death from the second round. Images are representative of $n = 3$ independent biological replicates. Scale bar, 3 μm. ORF, open reading frame. **b**, Growth curves for cells expressing an empty vector, SNIPE or PD-λ-3 and infected at different multiplicities of infection (MOI) of λ at $t = 0$ min. Optical density at 600 nm ($OD_{600}$) was used as a proxy for bacterial cell density and growth. The line shows the mean and the shaded region is the standard deviation; $n = 4$ independent biological replicates. **c**, Structure of SNIPE predicted by AlphaFold3, with colour-coded domains predicted by HHpred and DeepTMHMM. The N terminus, C terminus and E414A location are indicated. The start and end points of each domain are shown. **d**, Serial dilutions of λ spotted on lawns of cells expressing an empty vector or different SNIPE constructs. **e**, Serial dilutions of bacterial strains on plates with 0, 40 or 80 μM vanillate to induce empty vector or $P_{van}$–SNIPE constructs. **f**, Single-frame microscopy of cells expressing SNIPE–mCherry constructs and Gam–GFP, a marker of double-strand breaks. Images are representative of $n = 3$ independent biological replicates. Scale bar, 1 μm. **g**, Single-frame microscopy of cells expressing empty vector or SNIPE–GFP constructs. Nucleoids were stained with DAPI. Pearson's correlation coefficients ($R$) were calculated for each condition by comparing the GFP and DAPI fluorescence signals ($n > 100$ cells). Scale bar, 1 μm.

in SNIPE[15]. PhoA functions only in the periplasm and LacZα functions only in the cytoplasm, so the respective activities of these domains indicate the topological orientation of SNIPE. PhoA was functional only when inserted at the N terminus of SNIPE, and LacZα was functional only when inserted downstream of the transmembrane domain (Extended Data Fig. 1e). These data indicate that SNIPE is anchored in the inner membrane, with the DUF4041 and nuclease domains protruding into the cytoplasm.

To test whether the transmembrane, DUF4041 and GIY-YIG nuclease domains are necessary for defence against phage λ, we mutated each of these regions individually and performed plaquing assays. Substituting

a predicted catalytic residue in the nuclease domain (E414A) and deletion of the DUF4041 both abolished defence, indicating that these domains are essential for SNIPE function (Fig. 1d and Extended Data Fig. 1f). Attempts to clone SNIPE lacking the transmembrane domain (ΔTM) were unsuccessful, so we put SNIPE(ΔTM) under the control of a vanillate-inducible ($P_{van}$) promoter. Induced expression of this construct was highly toxic, and this toxicity was abolished by the E414A mutation (the toxicity was also partly reduced by ΔDUF4041, which is explored below) (Fig. 1e). These data indicated that SNIPE(ΔTM) might localize to and cleave host DNA. To test this model, we used Gam–GFP, a fluorescently tagged RecBCD inhibitor that localizes to double-strand

breaks in vivo[16]. We first confirmed that this marker was functional by expressing the restriction enzyme EcoRI fused to mCherry, which localized to and cut host DNA, as shown by Gam–GFP foci (Extended Data Fig. 1g). Similarly, we found that SNIPE(ΔTM)–mCherry localized to and cleaved host DNA (Fig. 1f). Consistent with the notion that the E414A mutation disrupts nuclease activity, no double-strand breaks were generated by SNIPE(ΔTM E414A)–mCherry.

To assess the subcellular localization of SNIPE and SNIPE(ΔTM) under the control of its native promoter, we first fused GFP to the C terminus of SNIPE and confirmed that this fusion protein retained robust defence against λ (Extended Data Fig. 2a). We found that SNIPE–GFP was uniformly localized to the cell membrane independent of phage infection (Fig. 1g). To visualize SNIPE(ΔTM)–GFP without the toxic effects of nuclease activity, we used the catalytically inactive variant SNIPE(ΔTM E414A)–GFP. This construct no longer localized to the cell membrane and instead associated with bacterial DNA, as judged by colocalization with the DAPI-stained nucleoid (Fig. 1g and Extended Data Fig. 2b). To confirm these results, we separated the cytoplasmic and membrane fractions of cell lysates and found by immunoblotting that SNIPE(E414A)–GFP was strongly enriched in the membrane fraction, whereas SNIPE(ΔTM E414A)–GFP was strongly enriched in the cytoplasm (Extended Data Fig. 2c). These results further indicate that the toxicity of SNIPE(ΔTM) stems from it localizing to and cleaving the host genome. By extension, these data indicate that wild-type SNIPE localizes to the inner membrane to help to sequester its nuclease activity and prevent autoimmunity.

Given that bacterial DNA probably contacts the cell membrane during processes such as chromosome replication and segregation[17], and that SNIPE is not intrinsically toxic to cells, we proposed that membrane-localized SNIPE does not cleave membrane-localized host DNA. To test this, we ectopically localized host DNA to the cell membrane by fusing two transmembrane domains from MalF to GFP–Fis, a fluorescently tagged DNA-binding protein. Strong expression of MalF(TM1-2)–GFP–Fis localized DAPI-stained DNA to the cell membrane and was toxic even in the absence of SNIPE (Extended Data Fig. 2d). Across a range of MalF(TM1-2)–GFP–Fis expression levels with varying degrees of toxicity, the presence of SNIPE generated no extra toxicity for cells. This result indicates that localization of host DNA to the cell membrane does not subject it to SNIPE-mediated cleavage, indicating that membrane-localized SNIPE exists in an auto-inhibited state.

To understand why removing the DUF4041 reduced the toxicity of SNIPE(ΔTM) (Fig. 1e), we examined the electrostatic surfaces of the predicted SNIPE structure and found that the DUF4041 domain contains a positively charged surface that might facilitate DNA binding (Extended Data Fig. 2e). To test this hypothesis, we compared the localization of SNIPE(ΔTM E414A)–GFP with and without the DUF4041 domain. Indeed, whereas SNIPE(ΔTM E414A)–GFP localized to the nucleoid, SNIPE(ΔTM E414A ΔDUF4041)–GFP exhibited diffuse cytoplasmic localization (Fig. 1g). Furthermore, we found that a version of SNIPE containing only DUF4041 and the downstream α-helix fused to GFP localized to the bacterial nucleoid (Extended Data Fig. 2f). These observations indicate that the DUF4041 promotes DNA binding by SNIPE(ΔTM). Collectively, our results support a model in which SNIPE contains a functional nuclease domain, a DUF4041 domain that facilitates DNA binding, and a transmembrane domain that anchors SNIPE to the inner membrane, preventing autoimmune cleavage of host DNA.

## SNIPE cuts phage DNA during injection

Given that SNIPE is a membrane-bound nuclease, we reasoned that SNIPE could directly cleave phage DNA during genome injection. We first confirmed that SNIPE does not affect the adsorption of phage λ (Fig. 2a), consistent with a previous study[13]. Next, we sought to visualize genome injection directly using a fluorescence assay in which λ carried a *parS* site and *E. coli* produced CFP-ParB[18]. After phage-genome injection, CFP-ParB oligomerized on *parS* sites and λ[parS] genomes appeared as fluorescent puncta. Consistent with previous work[18], CFP-ParB puncta appeared within ten minutes of λ[parS] infection and expanded during the course of phage-genome replication, which was followed by cell lysis (Fig. 2b,c and Supplementary Video 2). By contrast, the number of CFP-ParB puncta that appeared in SNIPE-expressing cells was reduced by about 30-fold, and there was a concomitant reduction in cell lysis. In rare cases in which a λ[parS] genome appeared in a SNIPE-containing cell, it went on to replicate and lyse the cell, indicating that if phages stochastically evade SNIPE at the cell membrane, their development proceeds unimpeded (Extended Data Fig. 3a,b and Supplementary Video 2). The SNIPE-dependent reduction in CFP-ParB puncta was not observed with the catalytically inactive E414A mutation, indicating that this reduction was dependent on nuclease activity (Fig. 2b,c).

To test directly whether phage DNA is cleaved by SNIPE, we adapted the classic Hershey–Chase experiment[19]. In this adaptation, we infected cells with λ containing ³²P-labelled DNA, collected cells shortly after genome injection and then measured the size of radiolabelled DNA fragments. Infection of empty vector cells yielded a clear ³²P-labelled band at the upper limit of size detection (4,000 base pairs), as would be expected for the injected, roughly 42,000-bp λ genome (Fig. 2d). This band was cleaved into a band of less than 100 bp following benzonase treatment, and this smaller band probably represents mononucleotides because [γ-³²P]-ATP formed a band of the same size (Extended Data Fig. 3c). Consistent with the ³²P-labelled band originating from injected phage DNA, it was not observed following infection of cells lacking the λ receptor (ΔlamB). In contrast to empty vector cells, infection of SNIPE-containing cells yielded a smear of DNA fragment sizes ranging from more than 4,000 bp to less than 100 bp, as well as the band that probably corresponded to mononucleotides. This profile of ³²P-labelled DNA cleavage was reversed in cells expressing SNIPE(E414A). Importantly, the amount of injected ³²P was similar across empty vector, SNIPE and SNIPE(E414A)-expressing cells, as measured by scintillation counting (Fig. 2d and Extended Data Fig. 3c). Taken together, our results strongly suggest that SNIPE cleaves phage DNA during the injection process.

We next investigated whether SNIPE could target a phage genome that was pre-existing in the cell. To this end, we introduced a plasmid expressing SNIPE or carrying an empty vector into a λ[parS] lysogen that produces CFP-ParB and encodes a temperature-sensitive cI repressor, which causes the λ[parS] prophage to enter the lytic cycle at high temperatures. Following heat shock, λ[parS] prophages were induced and then replicated, as manifest by CFP-ParB foci, followed by synchronous cell lysis (Fig. 2e and Supplementary Video 3). As predicted, SNIPE did not qualitatively affect the dynamics of CFP-ParB foci or cell lysis, nor did it affect the number of phage particles produced by prophage induction (Fig. 2f). These findings support the model that SNIPE cleaves phage DNA during, but not after, genome injection.

## SNIPE associates with ManYZ

We next wondered how SNIPE localizes to genome-injection sites. After adsorbing to the outer-membrane receptor LamB, phage λ requires the inner-membrane components of the mannose permease complex (consisting of ManY and ManZ) for genome injection, although the molecular details of this process remain poorly understood[20,21]. In principle, SNIPE could target phage DNA if it associates with LamB or ManYZ. After unsuccessful attempts to isolate λ escape mutants, we turned to a 'generalist' mutant of λ that evolved to infect ΔlamB and ΔmanYZ strains[22]. This phage has a mutation in the tail tip protein that allows it to bind to either LamB or an alternative outer-membrane protein, OmpF (Fig. 3a). This phage also has a mutation in the tape measure protein

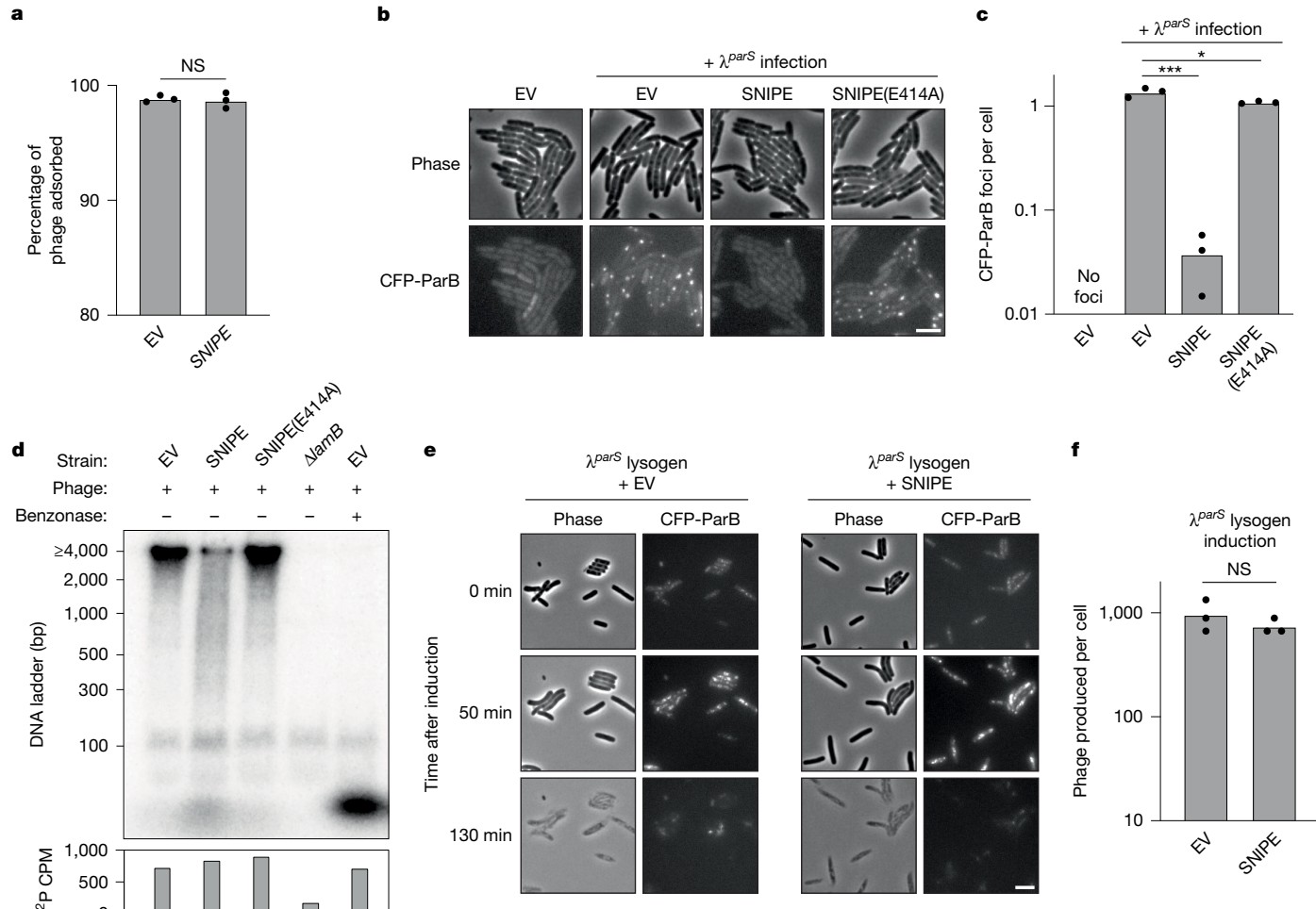

**Fig. 2 | SNIPE cleaves phage DNA during genome injection. a**, Adsorption assay with λ and cells containing SNIPE or an empty vector. Summary of $n = 3$ independent replicates. NS indicates $P = 0.76$ (unpaired two-tailed $t$-test). **b**, Microscopy of cells expressing CFP-ParB and an empty vector or different SNIPE constructs. Cells were infected with λ$^{parS}$ and imaged 5 min after genome injection. Scale bar, 2 μm. **c**, Quantification of CFP-ParB foci per cell from **b** and two more independent replicates. *$P = 0.03$, ***$P = 8.8 \times 10^{-5}$ (unpaired two-tailed $t$-tests). **d**, Various bacterial strains were infected with $^{32}$P-labelled λ and lysed 15 min after genome injection. Benzonase was added to one lysate sample. Lysates were subjected to electrophoresis through a polyacrylamide gel and imaged with a phosphor screen. Total $^{32}$P per sample was measured with a scintillation counter. Data are shown from one biological replicate. Data from one other independent biological replicate are shown in Extended Data Fig. 3c. **e**, λ$^{parS}$ lysogens expressing the heat-labile cI857 repressor, CFP-ParB and SNIPE or an empty vector were induced by heat shock at 42 °C. Time-lapse microscopy was used to monitor CFP-ParB foci dynamics and cell lysis. Scale bar, 2 μm. **f**, Quantification of plaque-forming units (PFU) per induced cell was done for the strains in **e** and two more independent replicates. NS, $P = 0.35$ (unpaired two-tailed $t$-test).

that allows it to circumvent a requirement for ManYZ, although the nature of this alternative genome-injection pathway remains unclear.

We first confirmed that the generalist λ mutant was able to infect Δ*lamB* and Δ*manYZ* strains, albeit with minor plaquing defects (Fig. 3b). Next, we reasoned that if SNIPE-mediated defence requires LamB or ManYZ, then shunting genome injection through one of the alternative pathways would circumvent defence. SNIPE provided robust defence against the generalist λ mutant in both wild-type and Δ*lamB* cells, demonstrating that LamB is not required for defence. By contrast, SNIPE-mediated defence was strongly reduced in Δ*manYZ*. These data indicate that, although the generalist λ mutant can use various genome-injection pathways, ManYZ is the preferred pathway through the inner membrane if provided. As such, the generalist λ mutant probably uses ManYZ in wild-type and Δ*lamB* cells, and is susceptible to SNIPE, whereas forcing this phage to use an alternative pathway in Δ*manYZ* cells largely circumvents SNIPE defence.

Our genetic analysis indicated that SNIPE requires the mannose permease complex to provide robust defence against λ. To assess interactions between SNIPE and ManYZ, we used biotin ligase proximity

labelling. This method involves fusing TurboID, which generates diffusible 5′-biotinyl-AMP that conjugates with nearby lysines, to a protein of interest and identifying the biotinylated proteins by streptavidin pull-down and mass spectrometry[23]. We first generated SNIPE–TurboID but found that it did not provide strong defence in the presence of biotin (Extended Data Fig. 4a). We therefore generated a plasmid expressing the *manXYZ* operon containing *TurboID-manZ*. This construct complemented λ plaquing defects on a Δ*manXYZ* strain and did not affect SNIPE-mediated defence, indicating that the TurboID-ManZ fusion was functional (Extended Data Fig. 4b,c).

To perform proximity labelling during phage-genome injection, we added wild-type λ and exogenous biotin to cells producing ManXY, TurboID-ManZ and SNIPE–GFP. At 15 min after infection, we lysed cells and used streptavidin pull-downs to isolate biotinylated proteins, which we identified by mass spectrometry. As expected, one of the most enriched proteins was AccB, which is the only protein that is naturally biotinylated in *E. coli* (Fig. 3c and Extended Data Fig. 4d). We also observed strong enrichment of ManX, a known interaction partner of ManYZ; ManY was not enriched, but harbours only one

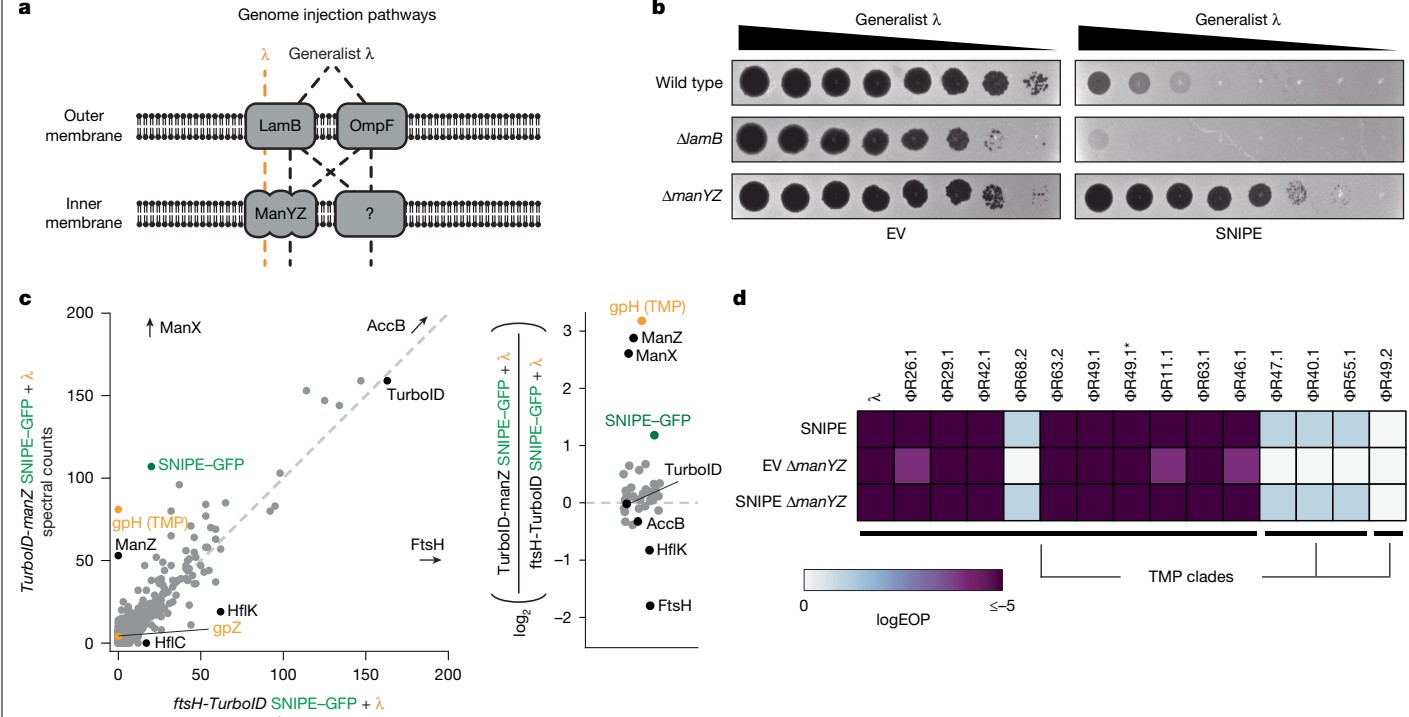

**Fig. 3 | SNIPE interacts with ManYZ. a**, Schematic of genome injection routes used by λ and the generalist λ mutant. **b**, Serial dilutions of the generalist λ mutant were spotted onto bacterial lawns with different genomic deletions and an empty vector or SNIPE-expressing plasmid. **c**, Proximity labelling was done with infected cells expressing SNIPE–GFP and *TurboID-ManZ* or *FtsH-TurboID*. Biotinylated proteins were enriched with streptavidin beads and quantified by mass spectrometry. The log$_2$ ratios of spectral counts for proteins with 50 or more spectral counts in at least one sample are shown on the right.

A pseudocount was added to each spectral count value to facilitate ratio calculations. Phage proteins are labelled in orange, SNIPE–GFP is labelled in green, and other proteins of interest are labelled in black. **d**, Efficiency of plaquing (EOP) data for λ and other temperate phages on wild-type or *ΔmanYZ* cells expressing SNIPE or harbouring an empty vector. Phages in a given TMP clade share more than 90% amino acid identity between their TMPs, and less than 20% identity with TMPs outside the clade.

cytoplasmic lysine that could be biotinylated[24]. Notably, SNIPE–GFP was also enriched, and the only phage protein that was robustly labelled by TurboID-ManZ during genome injection was the λ tape measure protein (TMP), also termed gpH (additional tail components gpJ and gpZ were weakly labelled in some TurboID-ManZ samples). To assess the specificity of these interactions, we fused TurboID to one of two other inner-membrane proteins, ProW and FtsH, and again performed proximity labelling during λ infection. We found that ManX, SNIPE–GFP and the λ tape measure protein were enriched only in the TurboID-ManZ samples (Fig. 3c and Extended Data Fig. 4c–e). These results indicate that ManYZ associates specifically with SNIPE and the TMP during genome injection.

To test whether SNIPE interacts with ManYZ independently of the TMP, we performed proximity labelling with TurboID-ManZ in the presence and absence of phage infection. Indeed, we observed strong enrichment of SNIPE–GFP in both cases, and similar results were obtained in a similar experiment involving untagged SNIPE (Extended Data Fig. 4f,g). These observations indicate that SNIPE and ManYZ interact before and during λ infection. Notably, this interaction did not impair the ManYZ-mediated transport of mannose, because SNIPE-containing cells retained the ability to metabolize mannose (Extended Data Fig. 4h,i). We also investigated whether the TMP interacts with ManYZ independently of SNIPE. During genome injection, we observed that TurboID-ManZ labelled the TMP at similar levels in the presence and absence of SNIPE (Extended Data Fig. 4j). Together, these data indicate that ManYZ interacts with both SNIPE and the TMP, independently of each other. By extension, our results indicate that SNIPE interacts with ManYZ before infection, positioning it to also associate with the TMP during infection and thereby target the incoming phage DNA for degradation.

If SNIPE interacts with ManYZ, then it should defend against other phages that also use ManYZ for genome injection. To test this prediction, we screened a panel of temperate phages that are related to λ[25]. Most of these phages were unable to infect a *ΔmanYZ* strain, indicating that they use ManYZ for genome injection (Fig. 3d). These same phages were highly susceptible to SNIPE-mediated defence. By contrast, phages that did not require ManYZ for infection were either unaffected or only modestly inhibited by SNIPE. This modest inhibition still occurred in *SNIPE ΔmanYZ* cells, indicating that SNIPE offers weak defence independently of ManYZ, an idea explored further below. Overall, these data indicate that SNIPE interacts with ManYZ to provide robust defence against phages that use this protein complex for genome injection.

Given our TurboID analyses, we also generated an alignment of the tape measure proteins encoded by each of the phages in this panel. These TMPs formed three distinct clades that strongly correlated with susceptibility to SNIPE and dependence on ManYZ (Fig. 3d and Extended Data Fig. 5a). Notably, in the clade that was highly susceptible to SNIPE and that required ManYZ for infection, there was one exception, the phage ΦR68.2. When comparing the ΦR68.2 TMP to other tape measure proteins in this clade, it harboured a region from amino acids 300 to 400 that differed substantially from the others (Extended Data Fig. 5b). This region overlaps with the A304V mutation in the tape measure protein of the generalist λ mutant that enables it to infect *ΔmanYZ* cells[22]. These results support the notion that phage genome injection using ManYZ substantially increases its susceptibility to SNIPE. However, SNIPE can provide some defence independently of ManYZ, as evidenced by the modest protection against phages ΦR40.1, ΦR47.1, ΦR55.1 and ΦR68.2 and by the reduced, but not fully abolished, defence that SNIPE provides against the generalist λ mutant in the *ΔmanYZ* background.

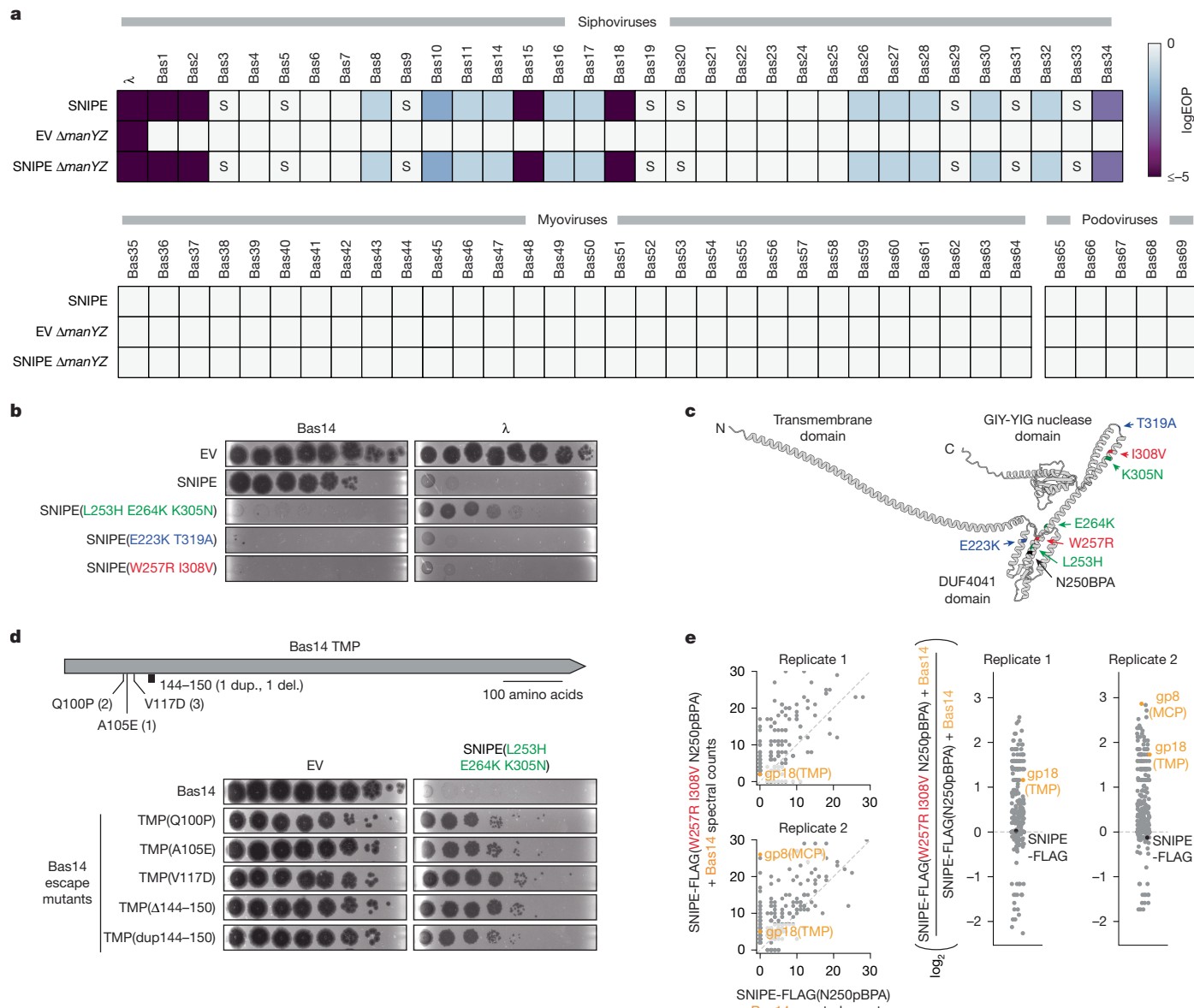

**Fig. 4 | SNIPE can target phages independently of ManYZ. a**, EOP data for λ and BASEL phages on wild-type or *ΔmanYZ* cells expressing SNIPE or harbouring an empty vector. S indicates smaller plaque sizes. **b**, Bas14 and λ were spotted onto lawns of *E. coli* expressing an empty vector, SNIPE or the indicated SNIPE mutants. **c**, Mutations that enhance SNIPE-mediated defence against Bas14 are highlighted on the structure of SNIPE predicted by AlphaFold3. The N250 residue replaced with pBPA in **e** is also indicated. **d**, Summary of identified escape mutations, all of which map to the Bas14 TMP. The number of independently isolated escaper plaques with a given genotype is shown in parentheses. Bas14 and Bas14 escape mutants were spotted onto lawns of empty vector and

SNIPE(L253H E264K K305N)-expressing cells; dup., duplication; del., deletion. **e**, UV-induced crosslinking of Bas14-infected cells expressing SNIPE-FLAG(W257R I308V N250pBPA) or SNIPE-FLAG(N250pBPA). These proteins and crosslinked products were pulled down with anti-FLAG beads and quantified by mass spectrometry. Scatterplot data are zoomed in to visualize proteins with low spectral counts; unzoomed data are shown in Extended Data Fig. 8b. The log₂ ratios of spectral counts are shown on the right. A pseudocount of 0.5 was added to each spectral count value to facilitate ratio calculations. Phage proteins are labelled in orange, and SNIPE-FLAG is labelled in black.

## SNIPE defends independently of ManYZ

To further test the determinants of SNIPE-mediated defence, we screened the BASEL collection of virulent phages for their susceptibility to SNIPE and their dependence on ManYZ[26]. None of the myoviruses or podoviruses in the BASEL collection was susceptible to SNIPE defence (Fig. 4a). However, most of the siphoviruses were weakly targeted by SNIPE, with a subset targeted more robustly. Surprisingly, none of these phages showed plaquing defects on *ΔmanYZ*, and SNIPE-mediated defence was unaffected in *ΔmanYZ* cells. These data indicate that SNIPE offers broad defence against most siphoviruses independently of ManYZ.

As noted above, SNIPE–TurboID was non-functional and thus could not be used to examine its interactions with proteins from these other phages. Thus, to probe the basis of SNIPE's broad defence against sipho-viruses, we used error-prone PCR to create a pool of cells harbouring mutagenized SNIPE and challenged it with phage Bas14. Although wild-type SNIPE provided approximately one log-fold protection against Bas14, three isolated SNIPE mutants strongly enhanced this defence, with each offering 6–7-log-fold protection (Fig. 4b). Each of these SNIPE mutants had at least two point mutations that mapped to the DUF4041 and the downstream α-helix (Fig. 4c). Notably, cloning single mutations de novo revealed that each mutation identified in the DUF4041 domain was sufficient to strongly enhance defence against Bas14, whereas

mutations in the downstream α-helix only partly enhanced this defence (Extended Data Fig. 6a,b). We suspect that these mutations were recovered together owing to the extreme selective pressure imposed by Bas14 infection in the original screening step. Given that Bas14 does not require ManYZ for entry, we inferred that these mutations in the DUF4041 domain enhanced the ManYZ-independent mechanism used by SNIPE.

To test whether the SNIPE mutants we isolated had improved defence generally or only against specific phages, we screened these mutants against λ and the siphoviruses in the BASEL collection. All three SNIPE mutants enhanced defence against only Bas14–18, which constitutes a clade of highly related phages (Extended Data Fig. 6c). This finding indicated that these SNIPE mutants enhance binding to part of the genome-injection apparatus used by Bas14–Bas18. To identify the basis of this enhanced defence, we selected Bas14 escapers on the SNIPE mutant harbouring the substitutions L253H, E264K and K305N. Strikingly, the only mutations found in escape phages mapped to the Bas14 gene that encodes the TMP (Fig. 4d). Three of these mutations were substitutions (Q100P, A105E and V117D), one was a deletion of amino acids 144–150, and one was a duplication of amino acids 144–150. All the escape mutants also provided some degree of escape against the other SNIPE mutants (E223K T319A and W257R I308V), although they were slightly more susceptible to wild-type SNIPE for reasons that remain unclear (Extended Data Fig. 7a,b). Notably, Bas14–Bas18 have TMPs that are more than 90% identical to each other and less than 20% identical to any other TMP in the collection (Extended Data Fig. 6c), and the region of the TMP that was mutated in Bas14 escapers (amino acids 100–150) had more than 85% sequence identity only in the Bas14–Bas18 clade (Extended Data Fig. 7c). Thus, Bas14 can escape the robust defence of SNIPE mutants by mutating a conserved region of its tape measure protein.

Next, we wondered how these SNIPE mutants provide enhanced defence in a manner that is dependent on the TMP. One possibility is that SNIPE binds weakly to diverse siphovirus tape measure proteins, and the SNIPE mutants strengthen this interaction for phages in the Bas14–Bas18 clade. Alternatively, the SNIPE mutants might enhance binding to an inner-membrane protein used by the Bas14–Bas18 clade, and Bas14 escaper mutations may switch to using a different inner-membrane protein. To test the latter possibility, we performed transposon-insertion sequencing (Tn-Seq)[27]. Specifically, we used λ, Bas14 or the Bas14 TMP(A105E) escape isolate to infect pools of cells in which barcoded transposons are inserted throughout the genome, such that most cells will die from infection, but cells harbouring a transposon in a gene necessary for phage infection will survive. As expected, this screen identified ManY and ManZ as required for infection of λ (Supplementary Table 1). By contrast, no integral inner-membrane proteins were required for infection of either Bas14 or Bas14 TMP(A105E). Notably, one or both of these phages could require essential inner-membrane proteins, which cannot be identified by Tn-Seq. Nevertheless, we find it unlikely that relatively minor mutations in Bas14 escapers conferred a complete switch from one essential inner-membrane protein to another, and instead we favour a model in which these phages do not use a specific host inner-membrane protein for genome injection. By extension, our results indicate that the SNIPE mutants could enhance binding to the Bas14 TMP, which is disrupted in Bas14 escapers.

To test whether the SNIPE DUF4041 interacts physically with the Bas14 TMP, we turned to crosslinking with the unnatural amino acid *p*-benzoylphenylalanine (pBPA). In this approach, we replaced the SNIPE N250 codon with an amber codon and expressed a specialized tRNA synthetase that can incorporate pBPA at this site[28]. Given that N250 is adjacent to the W257R mutation that enhanced defence against Bas14, we reasoned that pBPA incorporated at this site might crosslink to the Bas14 TMP (Fig. 4c). First, we confirmed that SNIPE–FLAG(W257R I308V N250pBPA) was still able to defend against Bas14 (Extended Data Fig. 8a). Next, we infected this strain with Bas14, exposed cells to ultraviolet light to crosslink pBPA to nearby proteins, and used anti-FLAG pull-downs to

identify crosslinked proteins using mass spectrometry. The only phage protein recovered across both replicates was the tape measure protein (Fig. 4e, Extended Data Fig. 8b and Supplementary Table 2). By contrast, no phage proteins were detected under similar conditions with SNIPE–FLAG(N250pBPA), indicating that recovery of phage proteins was dependent on the W257R I308V mutations. Together, these results indicate that the W257R I308V mutations enhance binding to the Bas14 TMP. Therefore, our results support a model in which wild-type SNIPE provides broad defence against siphoviruses by weakly interacting with siphovirus tape measure proteins. This defence could then be augmented by SNIPE binding to an inner-membrane protein used by the phage, such as ManYZ (as with λ) or by a strengthened interaction between DUF4041 and a specific type of TMP (as with Bas14).

## SNIPE homologues have diversified domains

To understand how evolutionary pressures shape the functional domains of SNIPE, we analysed a set of previously identified SNIPE homologues[13]. Consistent with this previous study, 33% of well-sequenced bacterial clades harbour at least one SNIPE homologue, and several clades harbour SNIPE homologues in up to 10% of their genomes, indicating that SNIPE provides anti-phage defence in diverse species (Extended Data Fig. 9a). We curated this set of SNIPE homologues with the clustering algorithm MMseqs2 at 95% identity to remove highly similar sequences and better understand the sequence diversity of these homologues[29]; this clustering yielded a set of around 500 homologues distributed across many bacterial phyla (Extended Data Fig. 9b). The most highly conserved region of SNIPE was the GIY-YIG nuclease domain, supporting the notion that nuclease activity is a crucial feature of SNIPE (Fig. 5a). By contrast, the DUF4041 domain was less well conserved, and the N-terminal region had very low sequence conservation. When quantifying the length of each domain in each homologue, we also found that the N-terminal region had highly variable lengths compared with the other regions (Fig. 5b). The diversity of the N-terminal region could indicate that it functions as an adapter that influences the phage specificity of SNIPE homologues.

We next investigated whether transmembrane domains are common among the N-terminal regions of SNIPE homologues despite their low sequence conservation. Indeed, a deep learning model predicted that 59% of SNIPE homologues harboured one transmembrane domain and 7% harboured two transmembrane domains[14] (Fig. 5c). These transmembrane domains were predicted to reside in the first 70 amino acids of SNIPE homologues and ranged from 10 to 20 amino acids in length (Fig. 5d,e). HMMER analysis of these SNIPE homologues containing predicted transmembrane domains indicated that their N-terminal regions are often similar to known bacterial inner-membrane proteins, such as FtsB, PilO and PspB (Fig. 5c and Supplementary Table 3). Furthermore, some N-terminal regions were homologues of the phage P22 tail needle protein, which is implicated in cell-envelope penetration[30]. Thus, despite the low sequence conservation of N-terminal regions, most of these regions probably target SNIPE homologues to the cell membrane and could interact with host and/or phage proteins to enhance defence.

For the remaining 34% of SNIPE homologues that did not contain a predicted TM, we investigated whether the N-terminal regions of these homologues contained other features that could target them to the membrane. A combination of AlphaFold2, structural-homology searches and sequence-homology searches with HMMER indicated that SNIPE homologues lacking predicted transmembrane domains frequently harbour domains from proteins such as DivIVA and type III secretion system ATPases, which localize to bacterial membranes by recognizing high negative membrane curvature and binding to inner-membrane proteins, respectively (Fig. 5c and Supplementary Table 3). Strikingly, some of these SNIPE homologues are predicted to harbour N-terminal globular domains with structural and sequence homology to pleckstrin homology domains, which can localize to

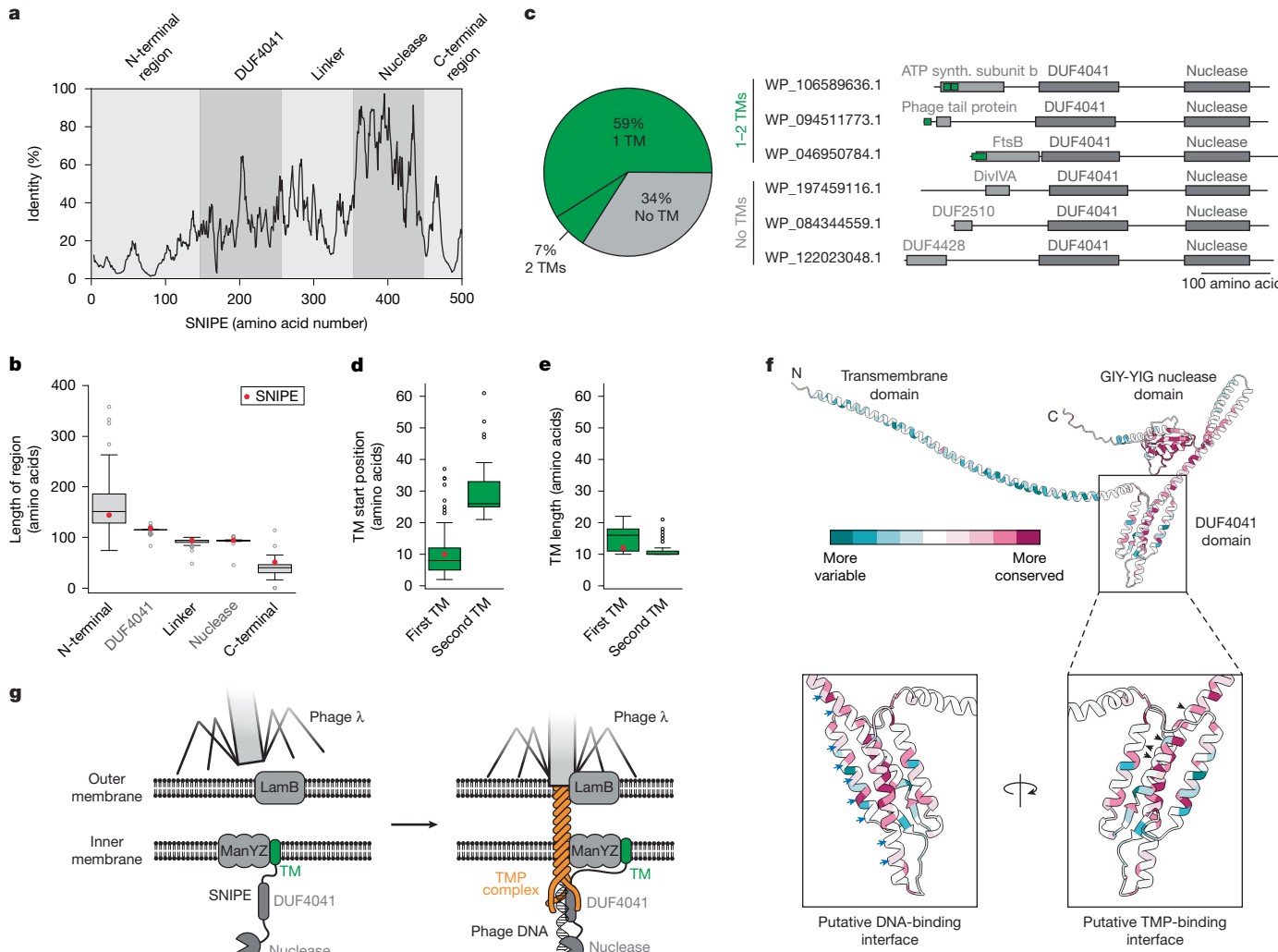

**Fig. 5 | SNIPE homologues harbour diverse N-terminal regions. a**, Sequence identity plot of SNIPE homologues, aligned and numbered relative to the amino acid positions of SNIPE. A five-residue rolling mean was applied. **b**, Length distributions of the extracted homologous sequences for each region of interest. The length of each region of SNIPE is shown as a red dot for reference. Centre line, median; box limits, upper and lower quartiles; whiskers, 1.5× the interquartile range; points, outliers; *n* = 474 SNIPE homologues. **c**, Percentage of SNIPE homologues that contain no, one or two transmembrane domains, as predicted by DeepTMHMM. Representative examples of SNIPE homologues with distinct N-terminal regions, predicted by HMMER, are shown as schematics. Synth., synthetase. **d**, Distribution of start positions for predicted transmembrane domains of SNIPE homologues. The length of the SNIPE

transmembrane domain is labelled as a red dot for reference. Centre line, median; box limits, upper and lower quartiles; whiskers, 1.5× interquartile range; points, outliers; *n* = 313 first TM domains, *n* = 33 second TM domains. **e**, As **d**, except for the length of the transmembrane domains. **f**, Evolutionary conservation of amino acids as estimated by ConSurf, overlaid onto the predicted structure of SNIPE. Insets indicate different interfaces of DUF4041. Black arrows indicate locations of SNIPE mutations that enhance defence against Bas14; blue arrows indicate positively charged amino acids that could facilitate DNA binding. **g**, Model for SNIPE-mediated defence against λ. The N-terminal region of SNIPE associates with ManYZ before phage infection. This interaction is maintained during phage infection, and DUF4041 binds to the TMP complex and phage DNA, positioning the nuclease domain to cleave the phage DNA.

membranes by binding to phospholipids (Extended Data Fig. 10a–e). Together, our analyses indicate that SNIPE homologues can localize to membranes by mimicking phage or bacterial proteins that harbour transmembrane domains or associate with membranes by other means.

Another crucial feature of SNIPE is the DUF4041 domain, which is required for defence, promotes DNA binding and probably interacts with a conserved feature of siphovirus tape measure proteins. To investigate DUF4041 evolution, we used ConSurf, an algorithm that measures the diversity of amino acid sequences across homologues of a protein of interest and maps those diversity scores onto the protein structure[31]. These analyses showed that the positively charged region of the DUF4041 domain is conserved relative to the rest of the domain, and that SNIPE homologues usually maintained positively charged residues in this region (Fig. 5f and Extended Data Fig. 10f). These data further support the notion that the positively charged interface binds to DNA, and that

this binding activity is broadly conserved. By contrast, the interface of the DUF4041 that probably binds to tape measure proteins exhibited more variability (Fig. 5f). In particular, the E223K and W257R mutations that enhanced binding to the Bas14 TMP are frequently found in SNIPE homologues (Extended Data Fig. 10f). These results indicate that the DUF4041 has been moderately diversified to strengthen binding to specific types of tape measure proteins. Taken together, our results indicate that the N-terminal region and the DUF4041 of SNIPE are under selective pressure to bind to distinct host and/or phage proteins to target different phages during genome injection.

## Discussion

Direct defence systems require the ability to discriminate between self and non-self, so the host is spared and the virus is targeted.

Most famously, CRISPR–Cas and restriction-modification systems identify non-self DNA based on sequence specificity and DNA modifications, respectively[32,33]. Here, our work reveals an alternative strategy, based on subcellular localization, with SNIPE specifically cleaving foreign DNA by localizing to the cell membrane and interacting with proteins associated with phage-genome injection (Fig. 5g).

Our proximity labelling studies indicate that SNIPE associates with ManYZ before an infection, as well as the tape measure protein during infection. The relative contributions of these two protein complexes to SNIPE defence is not yet clear. However, our finding that SNIPE can defend against a range of siphoviruses independently of ManYZ, along with the identification of SNIPE mutations that enhanced binding to the Bas14 TMP, indicates that the TMP interaction may be most important. Future structural studies are needed to understand these molecular interactions that underlie SNIPE-based defence. Such studies also promise to reveal how the nuclease domain of SNIPE is shielded from the host chromosome, which could contact the membrane at some locations[17].

Although most anti-phage defence systems act in the cytoplasm, there are now several immune systems that disrupt viral entry. Zorya localizes to phage genome injection sites, probably by sensing perturbations to the cell envelope, and is proposed to cleave phage DNA, although the exact mechanism remains unclear[34,35]. The Kiwa defence system also localizes to the cell membrane, where it could recognize membrane perturbations associated with genome injection, leading to DNA-binding activity that disrupts phage DNA replication[36,37]. Moreover, several super-infection exclusion mechanisms, used by phages to prevent immediate subsequent infection, act at the cell membrane[38,39]. Eukaryotes can block viral entry by using interferon-induced transmembrane proteins, which are thought to trap viruses in states of incomplete membrane fusion when they attempt to invade the host cytoplasm[40–43]. Together, these interferon-induced transmembrane proteins and bacterial systems such as SNIPE highlight cellular entry as a crucial vulnerability of the viral life cycle, and emphasize the importance of host defences that target these early steps of infection.

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

## Methods

### Growth conditions and strains

All the strains, phages, plasmids and oligonucleotides used in this study are listed in Supplementary Table 4. The sequences of all plasmids were confirmed by whole-plasmid sequencing. All bacterial strains were *E. coli* MG1655 derivatives unless noted otherwise. All λ strains in this study had mutations in the cI repressor and/or operator. Phages referred to as simply λ correspond to $\lambda_{vir}$, which cannot establish lysogeny owing to operator mutations and a frameshift in *cI*. The generalist λ mutant was derived from λ *cI*26, which has a frameshift in *cI* and is therefore incapable of establishing lysogeny, and $\lambda^{parS}$ harbours the *cI*857 mutation, which produces a temperature-sensitive repressor that is functional at 30 °C but not at 37 °C.

All the bacterial strains were grown in LB or M9 medium (6.4 g l$^{-1}$ Na$_2$HPO$_4$·7H$_2$O, 1.5 g l$^{-1}$ KH$_2$PO$_4$, 0.25 g l$^{-1}$ NaCl, 0.5 g l$^{-1}$ NH$_4$Cl, 0.1% casamino acids, 25 mM MgSO$_4$ and 0.1 mM CaCl$_2$) at 37 °C unless specified otherwise. Antibiotics were used at the following concentrations (liquid; plates): carbenicillin (50 µg ml$^{-1}$; 100 µg ml$^{-1}$); chloramphenicol (20 µg ml$^{-1}$; 30 µg ml$^{-1}$); and kanamycin (30 µg ml$^{-1}$; 50 µg ml$^{-1}$).

Phages were amplified by back-diluting overnight bacterial cultures to an optical density at 600 nm (OD$_{600}$) of around 0.05 in 1 ml LB medium and inoculating with a liquid phage stock at around 0.1 MOI or a single phage plaque. Phage amplification was done at 37 °C for 6–20 h. After this incubation, bacteria were pelleted by centrifugation and the supernatant was transferred to a new tube. This phage supernatant was either vortexed with an additional 100 µl of chloroform or passed through a 0.2 µm filter to remove remaining bacteria.

MacConkey agar plates were prepared using MacConkey agar powder (Difco) and 1% w/v mannose sugar. Strains were streaked out on these plates and grown at 30 °C for 12 h before imaging.

### SNIPE plasmids

SNIPE–GFP was generated by amplifying sfGFP-C1 (Addgene, 54579) with oDS6 and oDS7, and pKVS45-PD-λ-1 with oDS5 and oDS8, followed by Gibson assembly. P$_{van}$–SNIPE was generated by amplifying pKVS45-PD-λ-1 with oDS30 and oDS31 and the P$_{van}$ vector followed by Gibson assembly. The pBAD-SNIPE–mCherry, the parent plasmid for pBAD-SNIPE–mCherry with ΔTM, ΔDUF4041 and/or E414A mutations, was generated by amplifying pKVS45-PD-λ-1 with oDS473 and oDS499, mCherry-pBAD (Addgene, 54630) with oDS472 to oDS500, and a pBAD vector with oDS498 and oDS501, followed by Gibson assembly. SNIPE-TurboID-FLAG was generated by amplifying the TurboID-His6_pET21a plasmid (Addgene, 107177) using oDS665 and oDS668 and amplifying SNIPE-FLAG with oDS666 and oDS667 followed by Gibson assembly. SNIPE-FLAG was generated by amplifying pKVS45-PD-λ-1 with oDS3 and oDS4, followed by Gibson assembly.

To generate SNIPE(ΔTM) plasmids, parent plasmids were amplified with oDS47 and oDS48 followed by Gibson assembly. Similarly, SNIPE(ΔDUF4041) plasmids were generated by amplifying the parent plasmids with oDS51 and oDS52 followed by Gibson assembly. The E414A point mutation was introduced by amplifying parent plasmids with oDS92 and oDS93, followed by Gibson assembly. Other point mutations were generated by amplifying pKVS45-PD-λ-1 with the following primers: oDS923 and oDS924 (L253H); oDS925 and oDS926 (E264K); oDS927 and oDS928 (K305N); oDS929 and oDS930 (E223K); oDS931 and oDS932 (T319A); oDS933 and oDS934 (W257R); oDS935 and oDS936 (I308V); and oDS1075 and oDS1076 (N250amber/pBPA). These PCR products were then circularized by ligation with T4 DNA ligase.

Plasmids with different combinations of the DUF4041, linker and/ or catalytically inactive GIY-YIG nuclease domain fused to GFP were generated by amplifying SNIPE–GFP(E414A) with the following primers: oDS138 and oDS139 (DUF4041-linker-nuclease); oDS140 and oDS141 (linker-nuclease); and oDS142 and oDS143 (nuclease). These PCR products were then circularized by Gibson assembly. Two further

combinations were made by amplifying the DUF4041-linker-nuclease plasmid made above with the following primers: oDS144 and oDS145 (DUF4041); and oDS146 and 147 (DUF4041-linker). These PCR products were then circularized by Gibson assembly.

A PhoA–LacZα fusion protein was inserted into SNIPE(E414A) at different locations, as described previously[15]. Specifically, the PhoA–LacZα fusion protein was amplified from pKTop::tse5-CT (Addgene, 192955) with the following primers for insertion at different amino acids (a.a.) of SNIPE: oDS412 and oDS413 (SNIPE, a.a. 1); oDS416 and oDS417 (SNIPE, a.a. 30); oDS420 and oDS421 (SNIPE, a.a. 120); and oDS424 and oDS425 (SNIPE, a.a. 500). Furthermore, P$_{van}$–SNIPE(E414A) was amplified with the following primers: oDS411 and oDS414 (SNIPE, a.a. 1); oDS415 and oDS418 (SNIPE, a.a. 30); oDS419 and oDS422 (SNIPE, a.a. 120); and oDS423 and oDS426 (SNIPE, a.a. 500). The two PCR products for a specific insertion point were then assembled by Gibson assembly.

### Other plasmids

P$_{van}$-manXYZ was generated by amplifying the *manXYZ* operon from *E. coli* MG1655 with oDS171 and oDS172, and amplifying the P$_{van}$ vector with oDS170 and oDS173, followed by Gibson assembly. Next, P$_{van}$-manXYZ(TurboID-manZ) was generated by amplifying TurboID from TurboID-His6_pET21a with oDS810 and oDS811, and amplifying P$_{van}$-manXYZ with oDS809 and oDS812, followed by Gibson assembly.

To generate P$_{van}$-ftsH-TurboID, the P$_{van}$ vector was amplified with oDS883 and oDS891, ftsH was amplified from *E. coli* MG1655 with oDS892 and oDS893, and TurboID was amplified from TurboID-His6_pET21a with oDS882 and oDS887, followed by Gibson assembly. To generate P$_{van}$-proW-TurboID, the P$_{van}$ vector was amplified with oDS883 and oDS897, proW was amplified from *E. coli* MG1655 with oDS898 and oDS899, and TurboID was amplified from TurboID-His6_pET21a with oDS882 and oDS887, followed by Gibson assembly.

To generate pBAD-EcoRI R–mCherry, EcoRI R was amplified from a the pOpen-EcoRI R plasmid (Addgene, 165504) with oDS505 and oDS908, mCherry was amplified from pBAD-SNIPE–mCherry with oDS909 and oDS910, and the pBAD vector was amplified with oDS504 and oDS911; these products were then Gibson assembled.

To generate P$_{van}$-malF(TM1-2)-GFP-fis, fis was amplified from *E. coli* MG1655 with oDS1115 and oDS1116, malF(TM1-2) was amplified from *E. coli* MG1655 with oDS149 and oDS1112, GFP was amplified from sfGFP-C1 (Addgene, 54579) with oDS1113 and oDS1114, and the P$_{van}$ backbone was amplified with oDS148 and oDS1117, followed by Gibson assembly.

### *E. coli* gene deletions

Δ*manYZ*, Δ*manXYZ* and Δ*lamB* strains were generated using the lambda red recombination protocol derived from ref. 44. In brief, pKD4 was amplified with oDS94 and oDS95 to generate a Δ*manYZ*::kan$^R$ product, oDS95 and oDS174 to generate a Δ*manXYZ*::kan$^R$ product, and oDS96 and oDS97 to generate a Δ*lamB*::kan$^R$ product. These products were electroporated into either *E. coli* MG1655 or ECOR13 that contained a plasmid that can express lambda red proteins (pKD46). Kan$^R$ colonies that contained putative recombination events between the kan$^R$ cassette and the gene or operon of interest were isolated and confirmed with junction PCR.

### Phage and bacterial spotting assays

To prepare plates for phage spotting assays, 30 µl of a bacterial overnight culture was added to 4 ml of molten LB + 0.5% agar, and this mixture was then poured onto LB + 1.2% agar plates. After the top agar solidified, tenfold serial dilutions of a phage stock were made and 2 µl of each dilution was pipetted onto the plate using a multichannel pipette. Plates were incubated at 37 °C for 8–20 h before imaging. Images are representative of at least two independent biological replicates. EOP values were assessed qualitatively, given that strong defence prevents the formation of individual plaques. For one exception (Extended Data Fig. 7b), we used the top agar overlay method with different strains

of interest and quantified plaques for three independent biological replicates.

For bacterial spotting assays, overnight cultures of bacteria were back-diluted to an $OD_{600}$ of 1, tenfold serial dilutions of this bacterial stock were made, and 3 µl of each dilution was spotted onto LB + 1.2% agar + 0 µM, 40 µM or 80 µM vanillate plates. Plates were incubated at 37 °C for around 12 h before imaging. Images are representative of three independent biological replicates.

### Growth curve assays

Overnight cultures were back-diluted to an $OD_{600}$ of 0.01 and grown to mid-log phase in LB medium at 37 °C, then back-diluted to an $OD_{600}$ of 0.1. Then 100-µL aliquots of these cell solutions were added to wells of a 96-well plate, and 10 µl of different phage dilutions was then added to generate different MOIs. Samples were incubated at 37 °C with orbital shaking on a plate reader (Biotek) for 8 h, with $OD_{600}$ measurements every 10 min.

For growth in minimal media with glucose or mannose, overnight cultures were back-diluted to an $OD_{600}$ of 0.01 and grown to mid-log phase in LB medium at 37 °C. Cells were then washed twice with 1 ml minimal media (6.4 g $l^{-1}$ $Na_2HPO_4$-$7H_2O$, 1.5 g $l^{-1}$ $KH_2PO_4$, 0.25 g $l^{-1}$ NaCl, 0.5 g $l^{-1}$ $NH_4Cl$, 25 mM $MgSO_4$ and 0.1 mM $CaCl_2$) and resuspended in 1 ml aliquots of minimal media and 0.2% (w/v) glucose or minimal media and 0.5% (w/v) mannose at a final $OD_{600}$ of 0.05. Samples were incubated at 37 °C with orbital shaking on a plate reader (Biotek) for 28 h, with $OD_{600}$ measurements every 20 min.

### Plasmid transformation efficiency assay

Chemically competent cells were prepared using the transformation and storage solution (TSS) method. In brief, overnight cultures were diluted to an $OD_{600}$ of 0.05 and grown to mid-log phase in LB medium at 37 °C. Cells were collected by centrifugation, resuspended in ice-cold TSS buffer, aliquoted and kept on ice. For transformation, equal volumes of competent cells were mixed with defined amounts of an empty vector plasmid (pDSS240) and incubated on ice for 30 min. LB medium was added and cells were recovered at 37 °C for 1 h before plating on selective LB agar. Colony-forming units were counted after overnight incubation, and transformation efficiency was calculated as colony-forming units per ng of input plasmid DNA.

### Microscopy

To prepare cells for microscopy, overnight cultures were back-diluted to an $OD_{600}$ of 0.05 and grown in LB medium at 37 °C until cells reached the mid-exponential phase (an $OD_{600}$ of 0.3–0.4) unless specified otherwise.

Cells were then concentrated to an $OD_{600}$ of 1.0–1.5 to have a high cell density for microscopy. LB + 1.5% Ultrapure Agarose (Invitrogen, 16500-100) was melted and 600 µl was added to a 22 mm × 22 mm number 1.5 coverslip (VWR) and an identical coverslip was immediately placed on top. After the agarose pad solidified, the top coverslip was removed and 0.2 µl of cells were spotted onto the pad. For a given experiment, multiple bacterial strains were spotted onto the same agarose pad at different positions. After the spots soaked into the agarose pad, a 50 mm × 22 mm number 1.5 coverslip (VWR) was placed on top of the pad. Samples were imaged with phase contrast and epifluorescence channels on a Zeiss Observer Z1 microscope with a 100×/1.4 oil-immersion objective lens and a Colibri illumination system. Samples were kept at 37 °C during imaging by using a XLmulti S incubator (Pecon) and Heating Unit XL and TempModule S (Zeiss). An Orca Flash 4.0 camera (Hamamatsu) and Metamorph (Molecular Devices) were used for imaging. Image analysis was done using Fiji (ImageJ). All samples in an experiment are shown with similar brightness unless otherwise noted.

For SNIPE–GFP localization experiments, mid-exponential-phase cells were incubated with 10 µg $ml^{-1}$ DAPI for 10 min before imaging. These cells were imaged with phase contrast, DAPI and FITC channels. Images are representative of three independent biological replicates.

Pearson's correlation coefficients were calculated in Fiji (ImageJ) using the Coloc 2 plugin, following thresholding of the phase channel, conversion to a mask and segmentation with the watershed algorithm.

For experiments with Gam–GFP, cells were grown to mid-exponential phase in M9 media + 1% glucose (to repress SNIPE–mCherry and EcoRI R–mCherry) + 10 µg $ml^{-1}$ aTc (to express Gam–GFP) + carbenicillin + chloramphenicol. Cells were then washed with and resuspended in M9 media + 0.25% arabinose (to express SNIPE–mCherry or EcoRI R–mCherry) and incubated at 37 °C for 40 min. DAPI was then added to a 10 µg $ml^{-1}$ final concentration and cells were incubated at 37 °C for another 10 min. These cells were then imaged on a M9 media + 1.5% agarose pad with FITC, rhodamine, DAPI and phase-contrast channels. Images are representative of three biological replicates. Maximum pixel-intensity values were quantified in Fiji (ImageJ) after thresholding the phase channel, converting it to a binary mask and segmenting cells using the watershed algorithm. Individual cells were then analysed with the Analyze Particles → Measure function.

For experiments involving infection of CFP-ParB cells with $\lambda^{parS}$ phage, cells were grown to mid-exponential phase in LB medium + 20 µM IPTG (to induce CFP-ParB expression) + carbenicillin + chloramphenicol at 37 °C. Cells were concentrated to an $OD_{600}$ of 1.5 in 100 µl of the same media and placed on ice for 5 min. The $\lambda^{parS}$ was then added at an MOI of around 6 and incubated with cells on ice for 30 min to facilitate adsorption but prevent genome injection. This phage and cell mixture was then spotted onto a cooled LB + 1.5% agarose pad with 20 µM IPTG, carbenicillin and chloramphenicol. This agarose pad was quickly transferred to the microscope, which maintained the samples at 37 °C and thus triggered genome injection. Samples were imaged in the phase contrast and CFP channels every 5 min for 150 min. Quantification of CFP-ParB foci per cell and cell death per infected cell was done manually. Images are representative of three biological replicates.

For experiments involving the induction of $\lambda^{parS}$ $cI857$ prophages by heat shock, lysogens containing this prophage were grown to mid-exponential phase in LB + 80 µM IPTG (to induce CFP-ParB) + carbenicillin + chloramphenicol at 30 °C to prevent prophage induction. This cell solution was transferred to a 42 °C heat block for 5 min to initiate prophage induction. Cells were then spotted onto a pre-warmed LB + 1.5% agarose pad with 80 µM IPTG, carbenicillin and chloramphenicol. This pad was transferred to the microscope, where it was maintained at 37 °C and imaged in the phase contrast and CFP channels every 5 min for 150 min. The remaining cells that were not spotted onto the agarose pad were incubated at 37 °C for 4 h, after which plaque-forming units were quantified by spotting assays. Comparison of plaque-forming units per $OD_{600}$ of cells that were initially induced yielded values shown in Fig. 2f. Images and phage-spotting assays are representative of three independent biological replicates.

To test for direct defence and abortive infection phenotypes, cells were grown to mid-exponential phase in LB medium + chloramphenicol, followed by the addition of λ at 2 MOI, which ultimately yielded approximately one phage infection event for every two cells. This phage and cell mixture was incubated at 37 °C for 8 min to allow phage adsorption, followed by two rapid washes in LB medium + chloramphenicol to remove any unadsorbed phage. This phage and cell mixture was then spotted onto an LB + 1.5% agarose pad with chloramphenicol and imaged in the phase contrast channel for 3 h at 10-min time intervals. The imaging started exactly 20 min after the phage and cells were first mixed. Images and videos are representative of three independent biological replicates.

For experiments involving MalF(TM1-2)-GFP-Fis, cells were grown to mid-log phase in LB medium, and 50 µM vanillate was added. This solution was incubated at 37 °C for 30 min. DAPI was then added to a 10 µg $ml^{-1}$ final concentration and cells were incubated at 37 °C for another 10 min. These cells were then imaged on an LB + 1.5% agarose pad with the FITC, DAPI and phase contrast channels. Images are representative of three independent biological replicates.

## PhoA–LacZα topology assay

Membrane topology assays were performed as previously described[15,45], using the *E. coli* DH5α strain, which is naturally Δ*phoA* and is capable of β-galactosidase α-complementation (provided by LacZα). Specifically, three independent overnight cultures per strain were back-diluted to an $OD_{600}$ of 0.05 in 3 ml LB medium + 200 μM vanillate to induce $P_{van}$–SNIPE constructs and grown to late log phase (an $OD_{600}$ of about 0.8). To test for PhoA activity, 1 ml of each sample was washed once in 1 ml P buffer (1 M Tris-HCl, pH 8.0, and 0.1 mM $ZnCl_2$) and resuspended in 1 ml P buffer. Next, 50 μl of chloroform and 50 μl of 0.1% SDS was added to each sample and vortexed for 5 s to permeabilize the cells. This solution was incubated at 30 °C while chloroform separated from the aqueous layer. Then 150 μl of the aqueous layer was transferred to a 96-well plate. The enzymatic reaction was initiated by adding 18 μl of *p*-nitrophenylphosphate solution (Sigma, N7653) to each sample. Samples were incubated at 30 °C with orbital shaking on a plate reader (Biotek) for 2 h, with $OD_{405}$ measurements every 2 min. $OD_{600}$ was also measured to calculate the number of cells in each sample.

In parallel, to test for LacZ activity, 1 ml of each late-exponential-phase culture described above was washed once with 1 ml of Z buffer (60 mM $Na_2HPO_4$, 40 mM $NaH_2PO_4$, 10 mM KCl, 1 mM $MgSO_4$ and 50 mM β-mercaptoethanol added on the day of the experiment) and resuspended in 1 ml Z buffer. Next, 50 μl of chloroform and 50 μl of 0.1% SDS were added to each sample and vortexed for 5 s to permeabilize the cells. This solution was incubated at 30 °C while chloroform separated from the aqueous layer. Then 150 μl of the aqueous layer was transferred to a 96-well plate. The enzymatic reaction was initiated by adding 18 μl of 4 mg ml⁻¹ o-nitrophenyl galactopyranoside (Sigma, N1127) to each sample. Samples were incubated at 30 °C with orbital shaking on a plate reader (Biotek) for 2 h, with $OD_{420}$ measurements every 2 min. The $OD_{600}$ was also measured to calculate the number of cells in each sample.

To measure the enzymatic activities of PhoA and LacZ, the difference in $OD_{405}$ or $OD_{420}$ measurements between 0 and 60 min was calculated and calibrated to the starting $OD_{600}$ of the sample to provide the enzymatic activity per cell. These analyses were done separately for the three independent biological replicates before we calculated the averages and standard deviations.

## Membrane fractionation and immunoblots

Overnight bacterial cultures were back-diluted in LB medium and grown to an $OD_{600}$ of 0.6 at 37 °C. Cells were pelleted by centrifugation and resuspended in Buffer A (150 mM NaCl, 50 mM HEPES, pH 7) with EDTA-free protease inhibitor (Roche), 5 μl (150 kU) Ready-Lyse (Biosearch Technologies), 5 μl (125 U) benzonase (Millipore) and 2 mM $MgCl_2$. For measurements of protein levels in lysates, this cell solution was lysed in 1× Laemmli buffer with gentle heating at 50 °C, followed by electrophoresis and immunoblots as described below. For membrane fractionation, the cell solution was incubated on ice for 30 min to facilitate degradation of the peptidoglycan layer and then lysed by sonication. Lysate was centrifuged at 10,000*g* to pellet debris, and the supernatant was transferred to a new tube and centrifuged at 140,000*g* to pellet membranes. The supernatant from this spin was saved as the cytoplasmic fraction. The membrane pellet was briefly rinsed with Buffer A and resuspended in 2 ml of Buffer A with a Dounce homogenizer.

Samples were normalized by protein concentration, mixed with 4× Laemmli buffer and centrifuged at 13,000*g* for 10 min to pellet any debris. Next, 25 μl of each sample was subjected to electrophoresis with a 4–20% polyacrylamide gel and transferred to a 0.2 μm polyvinylidene difluoride membrane. Anti-GFP (Invitrogen, A11120), anti-DnaK (AssayPro, 32857-05111) and anti-OmpC (Bioss Antibodies, bs20213R) antibodies were used at a final concentration of 1:1,000, and goat anti-mouse IgG conjugated to HRP (Thermo, 32430) or goat anti-rabbit IgG conjugated to HRP (Thermo 32460) was used at a final concentration of 1:10,000. SuperSignal West Femto Maximum Sensitivity Substrate (Thermo) was used to develop blots. Blots were imaged using a ChemiDoc Imaging system (Bio-Rad). Images shown are representative of two biological replicates.

## Adsorption assay

Overnight bacterial cultures were back-diluted to an $OD_{600}$ of 0.05 in 1 ml of LB medium and grown to an $OD_{600}$ of 0.5 at 37 °C. These cultures were infected with λ at an MOI of 0.1 and incubated at 37 °C for 15 min. Samples were then centrifuged at 10,000*g* for 3 min and the phage-containing supernatant was transferred to a new tube containing 100 μl chloroform, which was then vortexed. The number of plaque-forming units per microlitre was calculated using the top agar overlay method with *E. coli* MG1655. Percentage adsorption was determined relative to a simultaneous control sample that contained growth medium but no cells. Data represent the averages and standard deviations of three independent biological replicates.

## Injection of radiolabelled phage DNA

A protocol to generate λ with ³²P-labelled DNA was designed using information from refs. 19,46. First, low-phosphate H-media (20 mM KCl, 85 mM NaCl, 20 mM $NH_4Cl$, 1 mM $MgSO_4$, 50 μM $CaCl_2$, 70 mM sodium lactate, 0.2% glycerol, 0.05% bacto-peptone (Difco) and 0.05% bacto-casamino acids (Difco)) was prepared as a 2× concentrate and diluted in water and 4% agar to generate molten H media + 1.2% agar and molten H media + 0.5% agar. Next, 5 ml of H media + 1.2% agar was poured into a 60 mm petri dish and allowed to solidify. Then 3 ml of the molten H media + 0.5% agar was mixed with 15 μl of Δ*dam E. coli* overnight culture (to prevent Dam methylation of DNA in this strain, the purpose of which is described below), 10,000 plaque-forming units of λ and 300 μCi ³²P (Revity). This mixture was immediately plated onto the H media 1.5% agar plate, which was incubated at 37 °C overnight. This protocol produced confluent lysis of the *E. coli*, which enabled recovery of much higher phage titres than similar attempts at phage amplification in liquid H media.

To recover ³²P-labelled phage, 1 ml of FM buffer (20 mM Tris-HCl, pH 7.4, 100 mM NaCl and 10 mM $MgSO_4$) was gently added to the top of the plate and incubated for 3 h at room temperature. This solution was then aspirated off of the plate and transferred to a microfuge tube. RNaseA was added at 10 μg ml⁻¹ and incubated for 5 min at room temperature to degrade the ³²P-labelled RNA. This solution was centrifuged at 10,000*g* to remove bacterial debris, transferred to a new tube with 10% volume chloroform, vortexed and centrifuged again. The supernatant was transferred to a 0.5 ml 50 kDa MWCO spin concentrator (Amicon) and repeatedly spin concentrated and resuspended in FM buffer to quickly remove the excess ³²P present in the media. This protocol typically yielded 200 μl of 10⁶ plaque-forming units of λ per microlitre.

To infect cells with ³²P-labelled λ, overnight cultures of different *E. coli* strains, which contained functional Dam and thus methylated their DNA, were back-diluted to an $OD_{600}$ of 0.1 in 1 ml of LB medium and grown at 37 °C until they reached an $OD_{600}$ of 0.4. These cell solutions were then centrifuged, resuspended in 100 μL of LB medium and placed on ice for 5 min. Next, the entire volume of radiolabelled phage was distributed equally across the bacterial samples, which yielded an MOI of approximately 0.1. These phage and cell mixtures were incubated on ice for 20 min to facilitate adsorption, then at 37 °C for 15 min to trigger genome injection. Cells were then spun down and resuspended in 1 ml of ice-cold PBS, and this wash step was repeated, effectively removing any unadsorbed phage particles. Cells were then spun down and resuspended in 25 μl ice-cold 1× rCut-Smart buffer + 0.5 μl (15 kU) Ready-Lyse (Biosearch Technologies) and incubated on ice for 5 min. This solution was flash-frozen with liquid nitrogen and thawed three times to lyse cells. Then 0.5 μl (10 U) of DpnI (NEB) was added to each sample and incubated at 37 °C for

5 min to degrade the Dam-methylated *E. coli* DNA and thus decrease the viscosity of the sample (notably, because phage were amplified in a *Δdam* strain, they did not have methylated DNA, and phage DNA was thus resistant to DpnI-mediated degradation).

Two controls were prepared at this time. First, one aliquot of empty vector cells that had been infected with $^{32}$P-labelled λ was treated with 1 μl (25 U) of benzonase (Millipore) and incubated at 37 °C for 5 min to degrade all the DNA in that sample. A second sample of empty vector cells that had been infected with $^{32}$P-labelled λ was mixed with a sample of SNIPE-expressing cell lysate that had not been infected with phage, and this mixture was incubated at 37 °C for 5 min. This effectively tested whether SNIPE is capable of cleaving phage DNA in cell lysates (that is, after genome injection had already occurred), which did not turn out to be the case.

To measure the length distributions of $^{32}$P-labelled DNA in these samples, these lysates were mixed with 4× Laemmli buffer and centrifuged at 13,000*g* for 10 min to pellet any remaining bacterial debris. Then 25 μl of each sample was loaded into a 4–20% pre-cast polyacrylamide gel (Bio-Rad). This gel was run at 100 V for 20 min and then 150 V for 25 min. The gel was incubated with a phorphorscreen and imaged using a Typhoon imager (GE Healthcare). Data for two independent biological replicates are shown.

### Proximity labelling with TurboID

Proximity-labelling assays were done as previously described[23], with some modifications. Overnight bacterial cultures were back-diluted to an $OD_{600}$ of 0.1 in 50 ml of LB medium + appropriate antibiotics with and without vanillate and grown at 37 °C to an $OD_{600}$ of 0.4. The $P_{van}$ promoter was used to express various constructs (TurboID-ManZ, FtsH-TurboID and ProW-TurboID). Leaky expression under this promoter in the absence of vanillate was sufficient to express TurboID-ManZ, but not FtsH-TurboID or ProW-TurboID. Therefore, 0 μM vanillate was used for some experiments (Extended Data Fig. 4b,f,g,j) and 5 μM vanillate was used for others (Fig. 3c and Extended Data Fig. 4c–e).

Following cell growth, these samples were then centrifuged, and cells were resuspended in 1.5 ml putrescine buffer (10 mM Tris-HCl, pH 7.4, 10 μM $MgCl_2$ and 10 mM putrescine dihydrochloride). Previous studies demonstrated that polyamines such as putrescine 'lock' λ DNA in the phage capsid, such that the genome-injection apparatus is probably able to assemble but DNA injection is not completed[47,48]; this phenomenon helped to stall otherwise transient genome-injection events to enable better proximity labelling of phage proteins. Biotin (dissolved in DMSO to 100 mM) was added at a 500 μM final concentration, λ was added at an MOI of 40, and these samples were placed at 37 °C for 15 min to facilitate biotinylation of phage and host proteins.

After this incubation step, samples were washed three times by centrifugation and resuspension of cells in 1 ml ice-cold putrescine buffer + 1 μl (25 U) benzonase (Millipore). Finally, cells were pelleted and resuspended in 1.5 ml RIPA buffer (50 mM Tris-HCl, pH 7.4, 150 mM NaCl, 1% Triton X-100, 0.5% sodium deoxycholate and 0.1% SDS) + EDTA-free protease inhibitor (Roche) + 1 μl (25 U) benzonase (Millipore) + 1 μl (30 kU) Ready-Lyse (Biosearch Technologies). Cells were lysed by sonication and then spun at 15,000*g* for 10 min to pellet cell debris. The supernatant was transferred to a fresh tube with 75 μl of streptavidin magnetic beads (Invitrogen, 65002) that had been pre-equilibrated in RIPA buffer. This mixture was incubated on an end-over-end rotor at 4 °C overnight.

Streptavidin beads were subsequently washed twice with each of the following buffers: RIPA buffer (1 M KCl, 100 mM $Na_2CO_3$), urea buffer (2 M urea, 10 mM Tris-HCl, pH 7.4) and PBS. On-bead reduction, trypsin digestion and liquid chromatography–tandem mass spectrometry (LC–MS/MS) were done by the MIT Biopolymers and Proteomics Core. Specifically, proteins were reduced with 10 mM dithiothreitol (Sigma) for 1 h at 56 °C, followed by alkylation with 20 mM iodoacetamide (Sigma) for 1 h at 25 °C in the dark. Digestion was done overnight using modified trypsin (Promega) in 100 mM ammonium bicarbonate (pH 8) at an enzyme-to-substrate ratio of 1:50. The digestion was terminated by the addition of formic acid (99.9%; Sigma). The resulting peptides were desalted using Pierce Peptide Desalting Spin Columns (Thermo) and subsequently lyophilized. Peptide separation was done on a PepMap RSLC C18 column (Thermo) over a 90-min gradient using reverse-phase high-performance liquid chromatography (Thermo, Ultimate 3000) followed by nano-electrospray ionization and analysis on an Orbitrap Exploris 480 mass spectrometer (Thermo). The mass spectrometer operated in data-dependent acquisition mode, with full-scan parameters set to a resolution of 120,000 across an *m/z* range of 375–1,600 and a maximum ion injection time of 25 ms. MS/MS acquisition was done for as many precursor ions as possible in a 2-s cycle, using a resolution of 30,000, a normalized collision energy of 28 and a dynamic exclusion window of 20 s. Peptides were mapped to proteins from *E. coli* MG1655, proteins from λ and proteins of interest, such as SNIPE–GFP and TurboID–ManZ. To compute ratios between spectral counts from different samples, a pseudocount was first added to each spectral count value. Ratios were computed only for proteins in which the number of spectral counts was 50 or more for at least one of the samples.

### Error-prone PCR and SNIPE mutant selection

SNIPE was mutagenized using error-prone PCR-based mutagenesis, as previously described[49]. Different regions of SNIPE (corresponding to amino acids 1–190, 191–330 and 331–500) were amplified using *Taq* polymerase (NEB), and 0.5 mM $MnCl_2$ was added to the reaction as the mutagenic agent. PCR products were treated with DpnI, gel extracted and separately cloned into the original pKVS45-PD-λ-1 plasmid using Gibson assembly, with the goal of creating three pools with mutations in these different regions. Gibson products were column purified and electroporated into *E. coli* MG1655. Around 10 million independent transformants were recovered for each pool, which were then combined to generate a mutagenized library of about 30 million SNIPE mutants.

To select SNIPE mutants with enhanced defence against Bas14, the SNIPE mutant library was grown to saturation overnight, 50 μl of this culture was mixed with Bas14 at an MOI of 1, and this solution was immediately plated on LB medium + 1.5% agar + chloramphenicol plates. After incubation overnight at 37 °C, Bas14-resistant colonies were picked, streaked for single colonies, grown into cultures and miniprepped, and the resulting plasmids were transformed into *E. coli* MG1655. Plasmids that conferred protection against Bas14 were sequenced to identify SNIPE mutations. Mutations were mapped onto the SNIPE structure predicted by AlphaFold3 using UCSF ChimeraX.

### Isolation and sequencing of phage escapers

Bas14 escapers were selected by mixing 30 μl of SNIPE(L253H E264K K305N) overnight culture and $10^9$ plaque-forming units of Bas14 with molten LB medium + 0.5% agar and using the top agar overlay method. Individual plaques were amplified in the SNIPE(L253H E264K K305N) strain and single plaques were isolated and amplified again to generate isogenic phage stocks.

Phage DNA was extracted by treating 200 μl of phage stock (over $10^7$ plaque-forming units per microlitre) with 0.2 U DNaseI and 0.05 mg ml$^{-1}$ RNaseA at 37 °C for 30 min. Then 10 mM EDTA was added to inactivate these nucleases. This solution was then incubated with Proteinase K at 50 °C for 30 min to disrupt phage particles and release DNA, which was then isolated by ethanol precipitation.

To prepare Illumina sequencing libraries, 200 ng of phage DNA was sheared in a Diagenode Bioruptor 300 sonicator water bath for 20 × 30 s cycles at maximum intensity. Sheared gDNA was prepared for Illumina sequencing using the NEBNext Ultra II DNA library preparation kit and sequenced on an Illumina MiSeq at the MIT BioMicroCenter. Illumina reads were assembled to the Bas14 reference genome using Geneious Prime 2025.3.

## Tn-Seq

Tn-Seq of cells in the presence or absence of phage infection was done as previously described[27]. A pool of cells harbouring random transposon insertions[50], each with a unique barcode and a kan[R] cassette, was grown to late-log phase and aliquots containing around $10^7$ cells were prepared. Different phages were added to separate tubes at about 1 MOI, and LB medium was added to a no-phage control. These samples were then immediately plated on LB medium + kanamycin plates and grown overnight at 37 °C. Colonies were scraped off of the plates and gDNA from 1 $OD_{600}$ unit of cells per sample was extracted with a DNeasy Blood and Tissue Kit (Qiagen). Next, 85 ng of DNA from each sample was amplified with BarSeq primers (Supplementary Table 4) to specifically amplify transposon barcodes with flanking Illumina adapter and index sequences. PCR products were pooled, treated with DpnI, run on an agarose gel and gel extracted. The pooled library was sequenced on an Illumina MiSeq at the MIT BioMicroCenter.

To count the number of reads for each barcode, we used previously written scripts[50], available at https://bitbucket.org/berkeleylab/feba. We $\log_2$-normalized read counts in each condition by calculating for each read count, $\log_2(\frac{\text{read count}}{\sum_{\text{condition}}\text{read counts}} \times 10^6 + 1)$. Then, to identify enriched barcodes, for each phage we fit a linear regression between the $\log_2$-normalized read counts of the no-phage condition (independent variable) and the $\log_2$-normalized read counts of the phage condition (dependent variable). For each barcode, the residual from the linear regression represented its deviation from expectation. We then mapped barcodes to genes using previously described mapping information[50] and averaged residuals for each gene to get $\log_2$ enrichment scores.

## Unnatural amino acid crosslinking

Overnight bacterial cultures harbouring different SNIPE–FLAG constructs and a plasmid with a suppressor tRNA and specialized tRNA synthetase (pSup-pBpaRS-6TRN)[28] were back-diluted to an $OD_{600}$ of 0.05 in 2 l of LB medium + 1.2 mM pBPA and grown at 37 °C to an $OD_{600}$ of 0.4. Cells were then pelleted by centrifugation and resuspended in 25 ml of putrescine buffer (10 mM Tris-HCl, pH 7.4, 10 μM $MgCl_2$ and 10 mM putrescine dihydrochloride). Bas14 was then added at 30 MOI and samples were exposed to UV with a Evoluchem 365 nm LED (HepatoChem) for 15 min. Cells were then pelleted by centrifugation and resuspended in 25 ml ice-cold Buffer A (50 mM HEPES, pH 7.5, 150 mM NaCl) + EDTA-free protease inhibitor (Roche) + 1 μl (25 U) benzonase (Millipore). Cells were lysed with an LM20 Microfluidizer (Microfluidics) at 18,000 psi and cell debris was pelleted at 10,000*g* for 30 min. Next, 20% Anzergent 3-12 was added to the supernatant to a final concentration of 1% to solubilize the membrane proteins. This solution was then spun at 100,000*g* for 1 h to pellet unsolubilized membranes, and the supernatant was incubated with 75 μl of Pierce anti-FLAG magnetic beads (Thermo) overnight at 4 °C. Beads were washed three times with Buffer A + 150 mM NaCl + 0.1% Anzergent 3-12, then washed three times with Buffer A + 150 mM NaCl to remove any remaining detergent. On-bead reduction, trypsin digestion and LC–MS/MS were done by the MIT Biopolymers and Proteomics Core, as described above. Peptides were mapped to proteins from *E. coli* MG1655 and Bas14. To compute ratios between spectral counts from different samples, a pseudocount of 0.5 was first added to each spectral count value.

## SNIPE homologue analysis

SNIPE homologues analysed in this study were originally identified in ref. 13. The original set of 1,141 homologues was condensed with MmSeqs2 with a clustering threshold of 0.95 and an alignment coverage threshold of 0.9 to remove highly similar homologues. This condensed list of 512 homologues was then aligned and manually curated to remove homologues that were clearly truncated versions of other homologues

in the set, as would be expected from misannotated start codons. The resulting list comprised 474 SNIPE homologues and was used for the analyses in this study.

The N-terminal regions of SNIPE were extracted and used in a HHblits search for hits in the Pfam database (PFAM), conserved domain database (cdd) from NCBI and the Protein Data Bank (PDB). Hits with a *P* value of less than $10^{-5}$ were considered significant and used for further analysis. These hits were sorted into two sets, one from SNIPE homologues with 1–2 predicted TMs, and the other from SNIPE homologues that lacked predicted TMs. Because there were usually multiple significant hits for each SNIPE homologue and many homologues had similar hits, we identified the most common hit within a set, calculated the number of SNIPE homologues containing that hit, and then removed all SNIPE homologues that contained one of these hits from further analysis. This process was iterated multiple times to generate a list of common hits for SNIPE homologues lacking TMs, and a separate list of common hits for SNIPE homologues containing 1–2 TMs, which is shown in Supplementary Table 3.

To search for structural homologues of N-terminal regions of SNIPE homologues that lacked predicted TMs, AlphaFold2 was used to predict structures for this set of SNIPE homologues. Each predicted structure was manually examined for N-terminal regions that harboured globular domains. To identify structural homologues of these domains, FoldSeek was used to scan the PDB100 database (v. 20240101).

## Reporting summary

Further information on research design is available in the Nature Portfolio Reporting Summary linked to this article.

## Data availability

Sequencing data are available in the Sequence Read Archive under BioProject PRJNA1231458. Summaries of spectral read counts and raw data for MS/MS of biotinylated or crosslinked proteins were deposited under MassIVE and can be accessed under accession MSV000097285. All other data are available in the manuscript or supplementary materials. Source data are provided with this paper.

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

**Acknowledgements** We thank T. Zhang, C. Vassallo and D. Nguyen for comments on the manuscript; J. Meyer for sharing the generalist λ mutant; I. Golding for sharing λ*parS* and CFP-ParB strains; staff at the MIT BioMicroCenter for support with Illumina sequencing; staff at the MIT Biopolymers and Proteomics Core for assisting with mass spectrometry; A. Millman for help with generating the unrooted phylogenetic tree; and A. Caruso for reagents and advice on pBPA crosslinking. M.T.L. is an Investigator of the Howard Hughes Medical Institute. This work was also supported by a Helen Hay Whitney Fellowship awarded to D.S.S.

**Author contributions** D.S.S., I.J.R. and M.T.L. conceived the study and designed experiments. D.S.S. did all the wet-lab experiments and collected the data. D.S.S., C.R.D. and P.C.D. did

bioinformatic and computational analyses. D.S.S. curated and analysed the data. D.S.S. and M.T.L. prepared the figures and visualizations. D.S.S. wrote the original draft of the manuscript. All authors contributed to manuscript review and editing. M.T.L. supervised the project and acquired funding.

**Competing interests** The authors declare no competing interests.

**Additional information**
**Correspondence and requests for materials** should be addressed to Michael T. Laub.

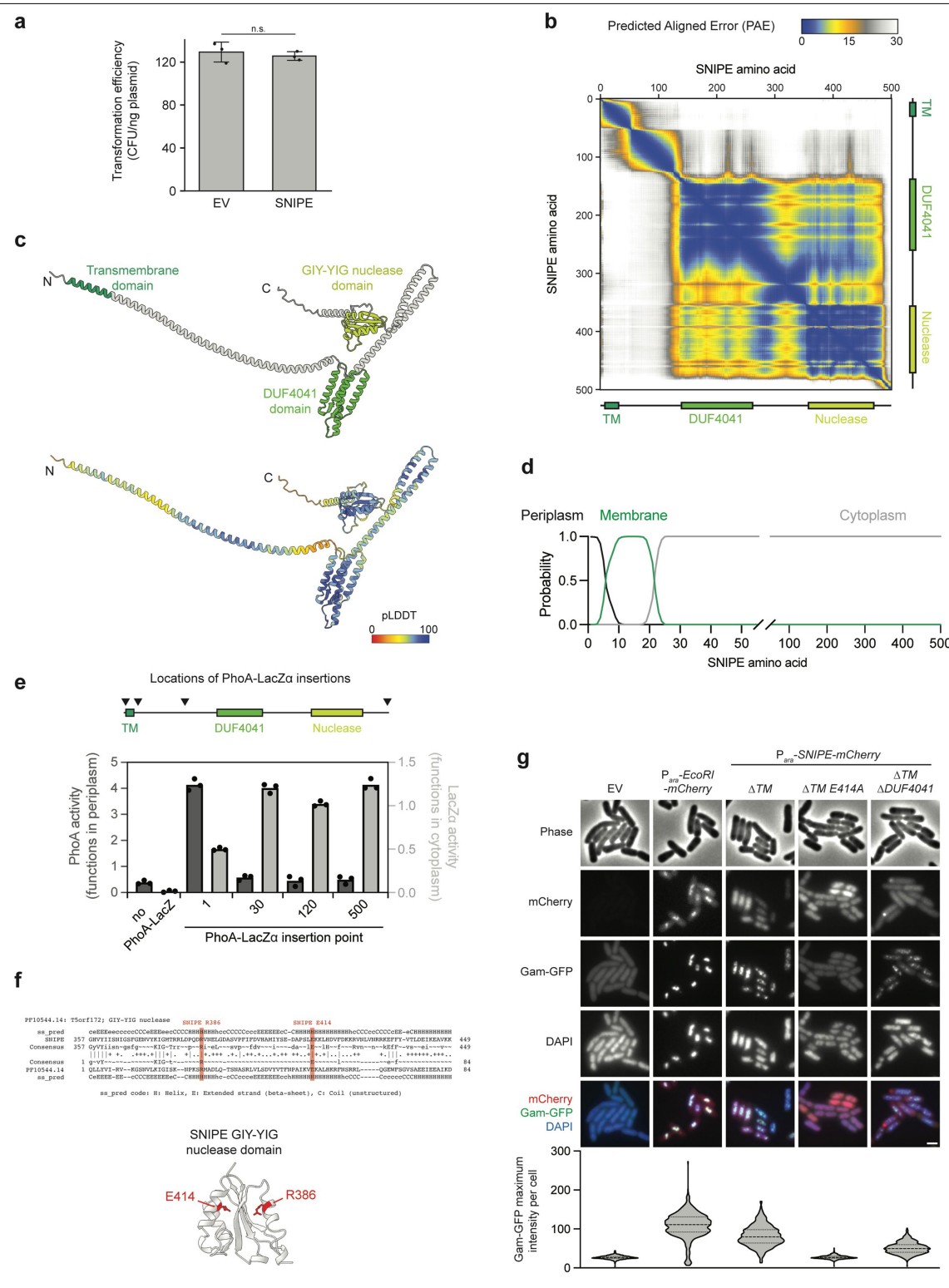

**Extended Data Fig. 1 | Predicted structure of SNIPE and membrane topology assay.** (**a**) Plasmid transformation efficiency assay using cells expressing an empty vector (EV) or SNIPE. An empty vector plasmid with a different antibiotic resistance marker was transformed into chemically competent cells. Colony forming units (CFU) were enumerated and used to calculate CFU/ng of empty vector plasmid. Summary of n = 3 independent biological replicates. n.s. indicates p = 0.58 (unpaired two-tailed t-test). (**b**) PAE plot for SNIPE structure predicted by AlphaFold3. (**c**) The structure of SNIPE predicted by AlphaFold3, as shown in Fig. 1c, with pLDDT scores shown on the bottom structure. (**d**) Predicted topology of SNIPE residues calculated by DeepTMHMM. (**e**) Insertion points of PhoA-LacZα in SNIPE(E414A) are shown in the schematic. PhoA and LacZ activities were calculated by permeabilizing cells in the presence of colorimetric substrates *p*-nitrophenylphosphate and o-nitrophenyl galactopyranoside, respectively. Summary of n = 3 independent biological replicates. (**f**) HHpred analysis of SNIPE shows that the nuclease domain is a member of the GIY-YIG nuclease family. The structure of the nuclease domain, as predicted by AlphaFold3, is shown on the bottom. Predicted catalytic residues are highlighted in red. (**g**) Single-frame microscopy of cells expressing an empty vector or P_ara-SNIPE-mCherry constructs, and Gam-GFP, a marker of double strand breaks. The nucleoid was stained with DAPI. A subset of these images is shown in Fig. 1f. Given that Gam-GFP foci often coalesce[16], the maximum Gam-GFP pixel intensity per cell is shown as a violin plot. Center dashed line, median; top and bottom dashed lines, upper and lower quartiles. n ≥ 431 cells per sample. Scale bar, 1 μm.

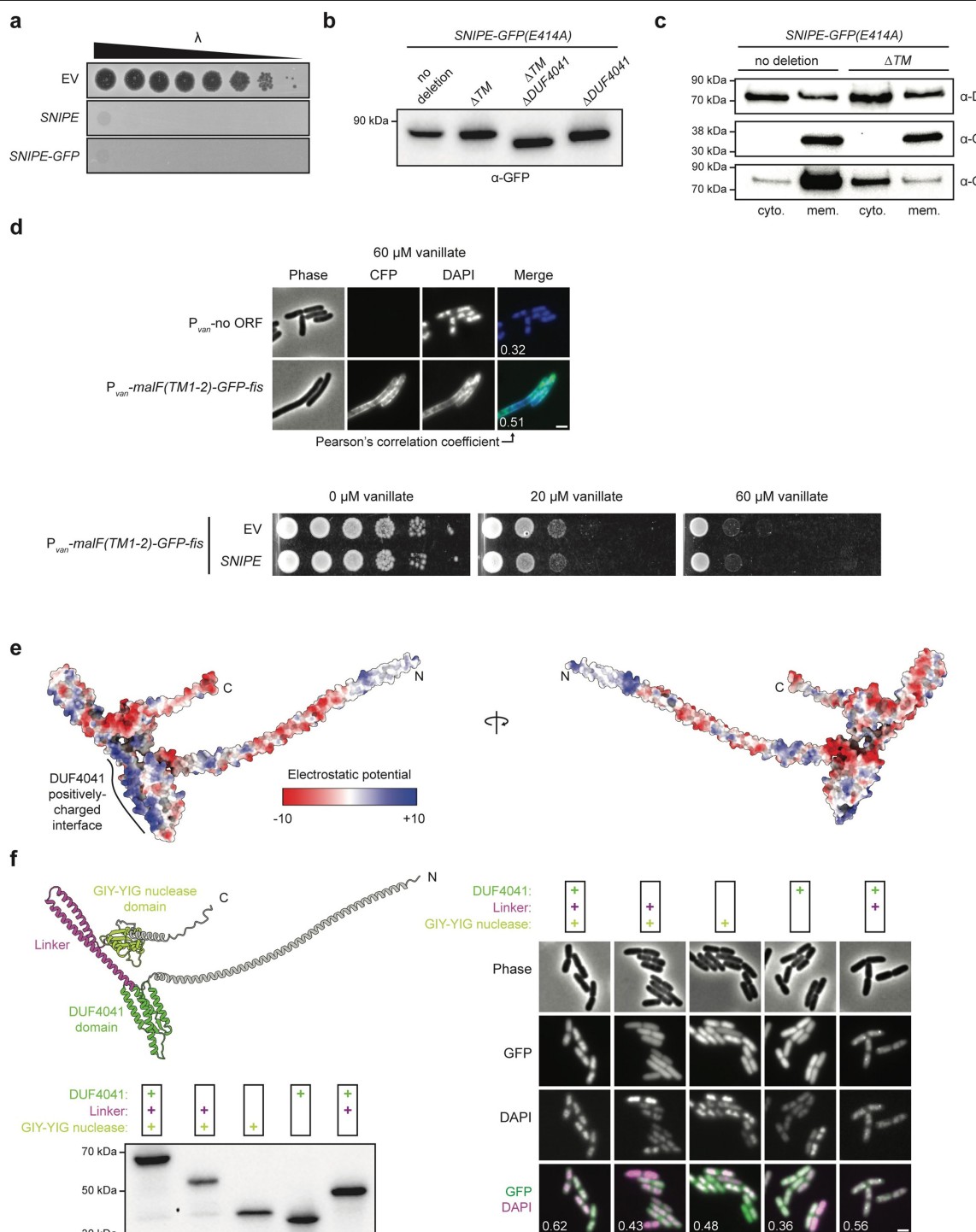

**Extended Data Fig. 2 | Localization and nuclease activity of fluorescently tagged SNIPE.** (**a**) Serial dilutions of λ were spotted onto lawns of cells expressing an empty vector (EV), SNIPE, or SNIPE-GFP. (**b**) Expression levels of different SNIPE-GFP(E414A) constructs as shown by immunoblots. The same membrane was stained with Coomassie as a loading control. The image is representative of n = 2 independent biological replicates. For gel source data, see Supplementary Fig. 1a. (**c**) Immunoblots of separated cytoplasm and membrane fractions isolated from cells expressing SNIPE(E414A)-GFP or SNIPE(ΔTM E414A)-GFP. Immunoblots for DnaK, which is mostly cytoplasmic[51], and OmpC, which is a membrane protein[52], are provided as controls. The anti-GFP membrane was stained with Coomassie as a loading control. Images are representative of n = 2 independent biological replicates. For gel source data, see Supplementary Fig. 1b. (**d**) Strains expressing MalF(TM1-2)-GFP-Fis or a negative control plasmid were imaged with fluorescence and phase microscopy. The nucleoid is stained

with DAPI. Scale bar, 2 μm. Additionally, toxicity of strains expressing MalF(TM1-2)-GFP-Fis and harboring SNIPE or an empty vector control was assessed with bacterial spotting assays on plates with 0, 20, or 60 μM vanillate. (**e**) Electrostatic surfaces of the AlphaFold3-predicted SNIPE structure. The positively charged surface of the DUF4041 is highlighted. (**f**) The structure of SNIPE predicted by AlphaFold3, with color-coded domains of interest. Different combinations of these domains were fused to GFP and imaged with fluorescence and phase microscopy. The nucleoid was stained with DAPI. All constructs that contained the nuclease domain harbored the catalytically inactive E414A mutation. Images are representative of n = 3 independent biological replicates. Scale bar, 2 μm. Expression levels of these constructs were assessed with immunoblots. The same membrane was stained with Coomassie as a loading control. Images are representative of n = 2 independent biological replicates. For gel source data, see Supplementary Fig. 1a.

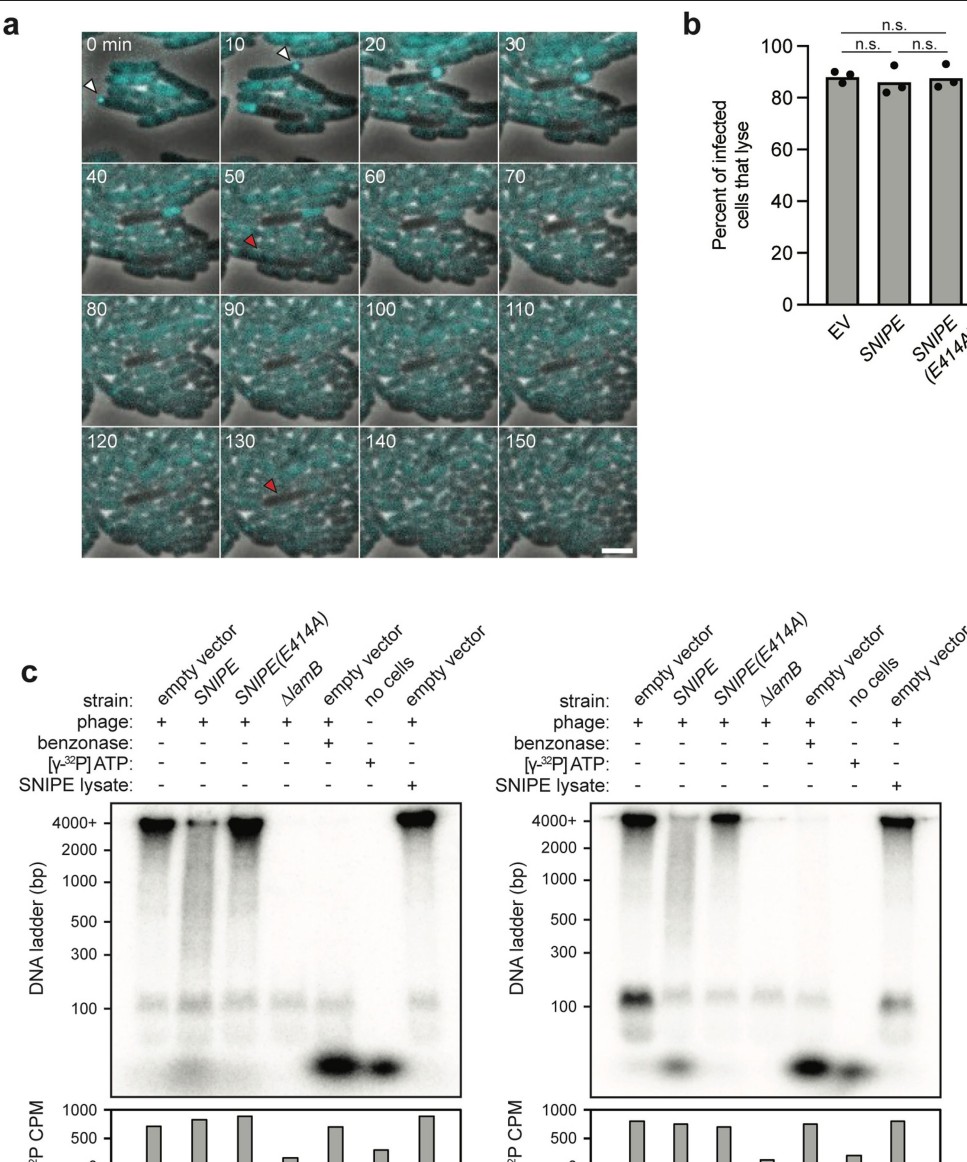

**Extended Data Fig. 3 | Cell lysis of λ infection events that evade SNIPE and SNIPE-mediated cleavage of phage DNA.** (**a**) Cells expressing SNIPE and CFP-ParB were infected with λ$^{parS}$ and imaged with time-lapse microscopy. The experiment is identical to Fig. 2b but a different field of view is shown for clarity in montage format. Phase and CFP channels are merged. White arrows indicate phage infection events. Red arrows indicate infected cells that lyse in the subsequent frame. Scale bar, 2 μm. (**b**) Cells which showed visible CFP-ParB foci during λ$^{parS}$ infection were tracked with time-lapse microscopy and the percent of these cells that lysed were quantified. Analysis was performed on samples shown in Fig. 2b-c and two additional independent replicates. n.s. indicates p > 0.5 (unpaired two-tailed t-tests). (**c**) Different bacterial strains were infected with $^{32}$P-labeled λ and lysed 15 min after genome injection. Benzonase was

added to one lysate sample. Lysates were subjected to electrophoresis through a polyacrylamide gel and imaged with a phosphorscreen. Total $^{32}$P per sample was measured with a scintillation counter. n = 2 indepedent biological replicates are shown. The replicate on the left is identical to Fig. 2d but shows two additional controls. One control was [γ−$^{32}$P] ATP in the absence of cell lysate and phage. In the other control, lysate from empty vector cells infected with $^{32}$P-labeled λ was mixed with uninfected SNIPE-expressing cell lysate to test if SNIPE can cleave phage DNA after lysis. The second independent biological replicate is shown on the right. The bands at ~50 and ~100 bp were considered as nonspecific background due to their presence in the ΔlamB strain. For gel source data, see Supplementary Fig. 1c.

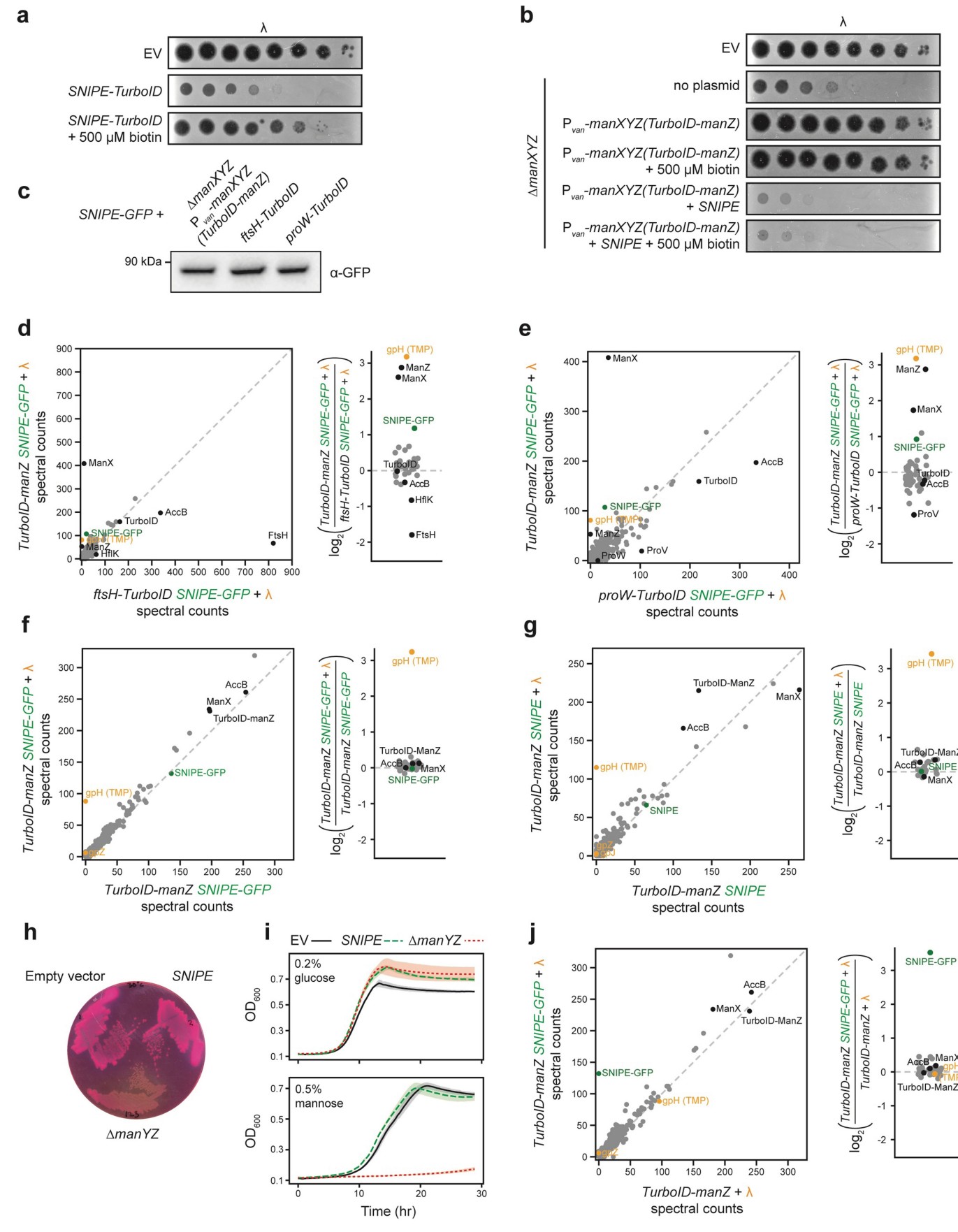

**Extended Data Fig. 4** | See next page for caption.

**Extended Data Fig. 4 | Proximity labeling with TurboID. (a)** Serial dilutions of λ were spotted onto lawns of cells expressing an empty vector or SNIPE-TurboID, in the presence or absence of 500 μM biotin. **(b)** Serial dilutions of λ were spotted onto lawns of cells expressing an empty vector or *ΔmanXYZ* cells expressing no plasmid, TurboID-ManZ, and/or SNIPE, in the presence or absence of 500 μM biotin. **(c)** Expression levels of SNIPE-GFP in different TurboID fusion protein backgrounds, as assessed with immunoblots. The same membrane was stained with Coomassie as a loading control. The image is representative of n = 2 independent biological replicates. For gel source data, see Supplementary Fig. 1a. **(d)** Proximity labeling was performed with infected cells expressing SNIPE-GFP and TurboID-ManZ or FtsH-TurboID. Biotinylated proteins were enriched with streptavidin beads and quantified by mass spectrometry. Log$_2$ ratios of spectral counts for proteins with ≥ 50 spectral counts in at least one sample are shown on the right. Phage proteins are labeled in orange, SNIPE is labeled in green, and other proteins of interest are labeled in black. Data is identical to Fig. 3c but zoomed out to permit visualization of all data points. **(e)** Same as **(d)**, but comparing infected cells expressing SNIPE-GFP and TurboID-ManZ or ProW-TurboID. **(f)** Same as **(d)**, but comparing infected versus uninfected cells expressing SNIPE-GFP and TurboID-ManZ. **(g)** Same as **(d)**, but comparing infected versus uninfected cells expressing SNIPE and TurboID-ManZ. **(h)** Empty vector, *SNIPE*, and *ΔmanYZ* colonies grown on MacConkey agar + 1% w/v mannose. Pink colonies indicate successful metabolism of mannose. **(i)** Growth curves of empty vector (EV), *SNIPE*, and *ΔmanYZ* cells grown on minimal media + 0.2% w/v glucose or 0.5% w/v mannose. Line = mean; shaded region = standard deviation. n = 3 independent biological replicates. **(j)** Same as **(d)**, but comparing infected cells expressing TurboID-ManZ versus cells expressing TurboID-ManZ and SNIPE-GFP.

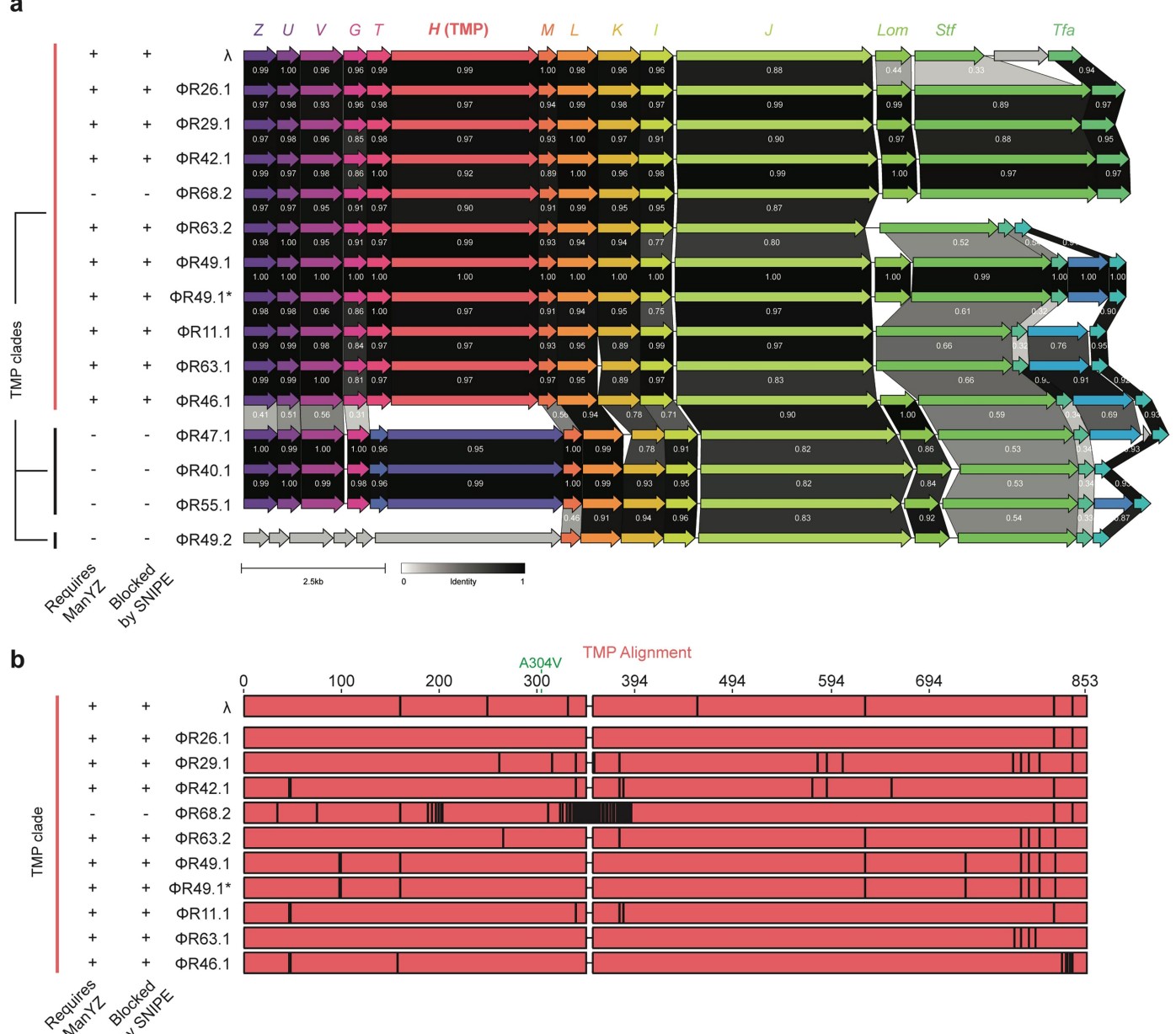

**Extended Data Fig. 5 | Alignments of tail genes from λ and a temperate phage panel. (a)** Tail gene clusters visualized with the CAGECAT clinker web server[53]. Sequence identity between genes (on a scale of 0 to 1) is shown. λ genes are labeled for reference. The propensity of each phage to be targeted by SNIPE and use ManYZ for genome entry is derived from data in Fig. 3d. Phages in a given TMP clade share > 90% amino acid identity between their TMPs, and <20% identity with TMPs outside of the clade. **(b)** Alignment between TMPs of λ and other phages with the same TMP clade. Black lines indicate disagreements with the consensus sequence of the alignment. A304V indicates the mutation that is present in the generalist λ mutant, which allows it to circumvent a requirement for ManYZ. Amino acid numbers in the λ TMP are shown for reference. The propensity of each phage to be targeted by SNIPE and use ManYZ for genome entry is derived from data in Fig. 3d.

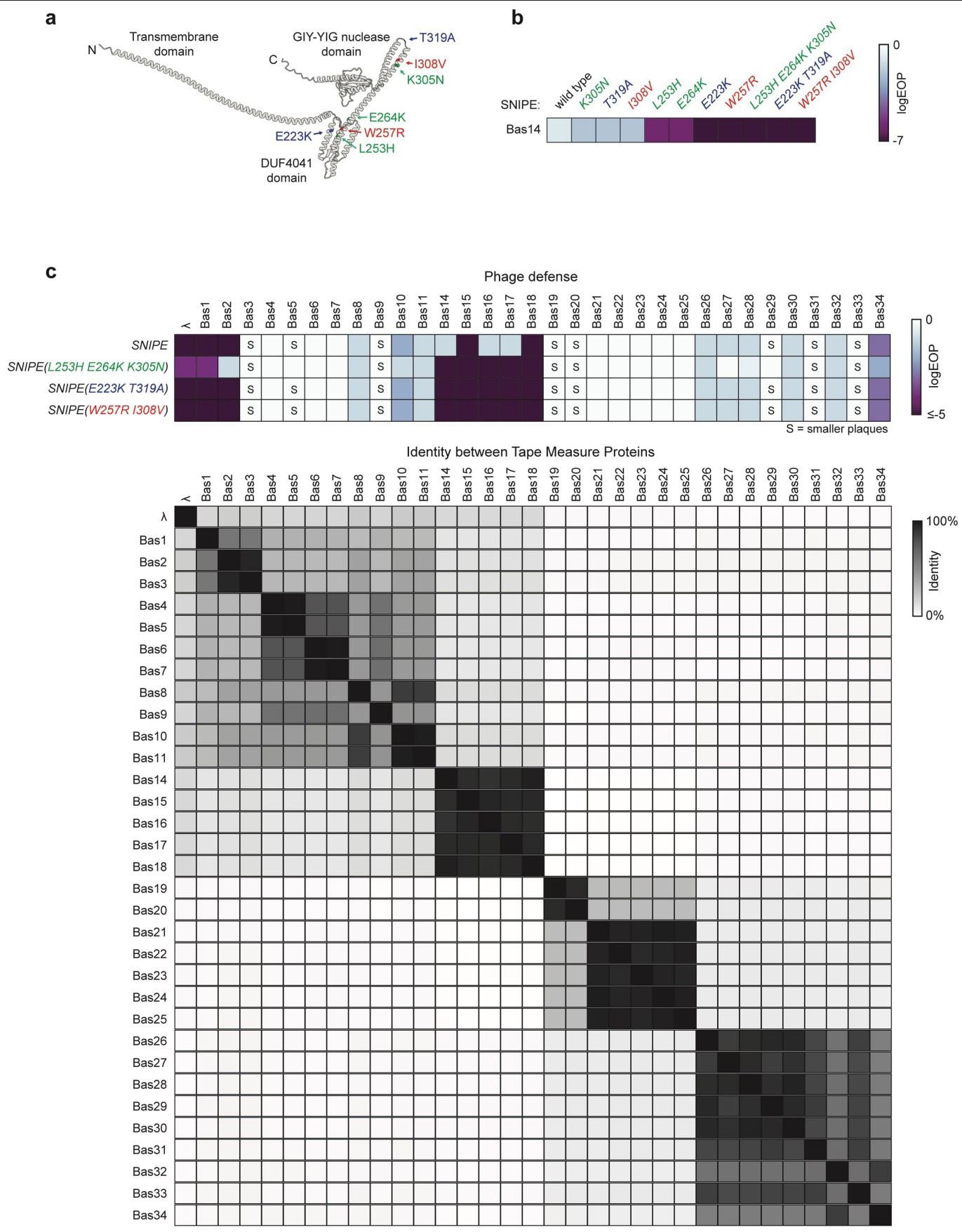

**Extended Data Fig. 6 | SNIPE mutations enhance defense against the Bas14-18 clade.** (**a**) Mutations that enhance SNIPE-mediated defense against Bas14 are highlighted on the structure of SNIPE predicted by AlphaFold3. The same schematic is shown in Fig. 4c and shown here for reference. (**b**) EOP data for Bas14 on SNIPE and different SNIPE mutants. (**c**) EOP data for λ and BASEL siphoviruses on SNIPE and different SNIPE mutants. Smaller plaque sizes are indicated with an 'S'. Amino acid sequence identity between TMPs of these phages is shown in matrix format.

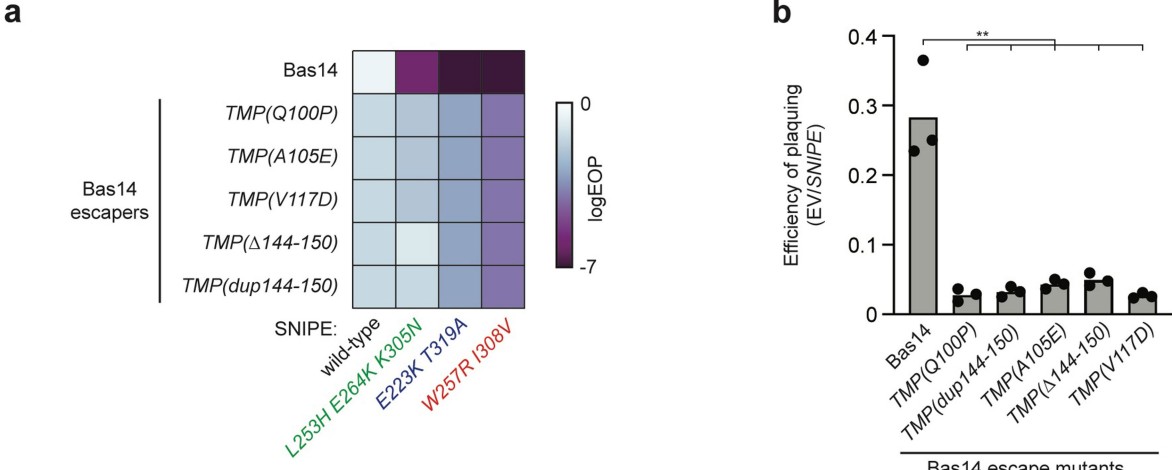

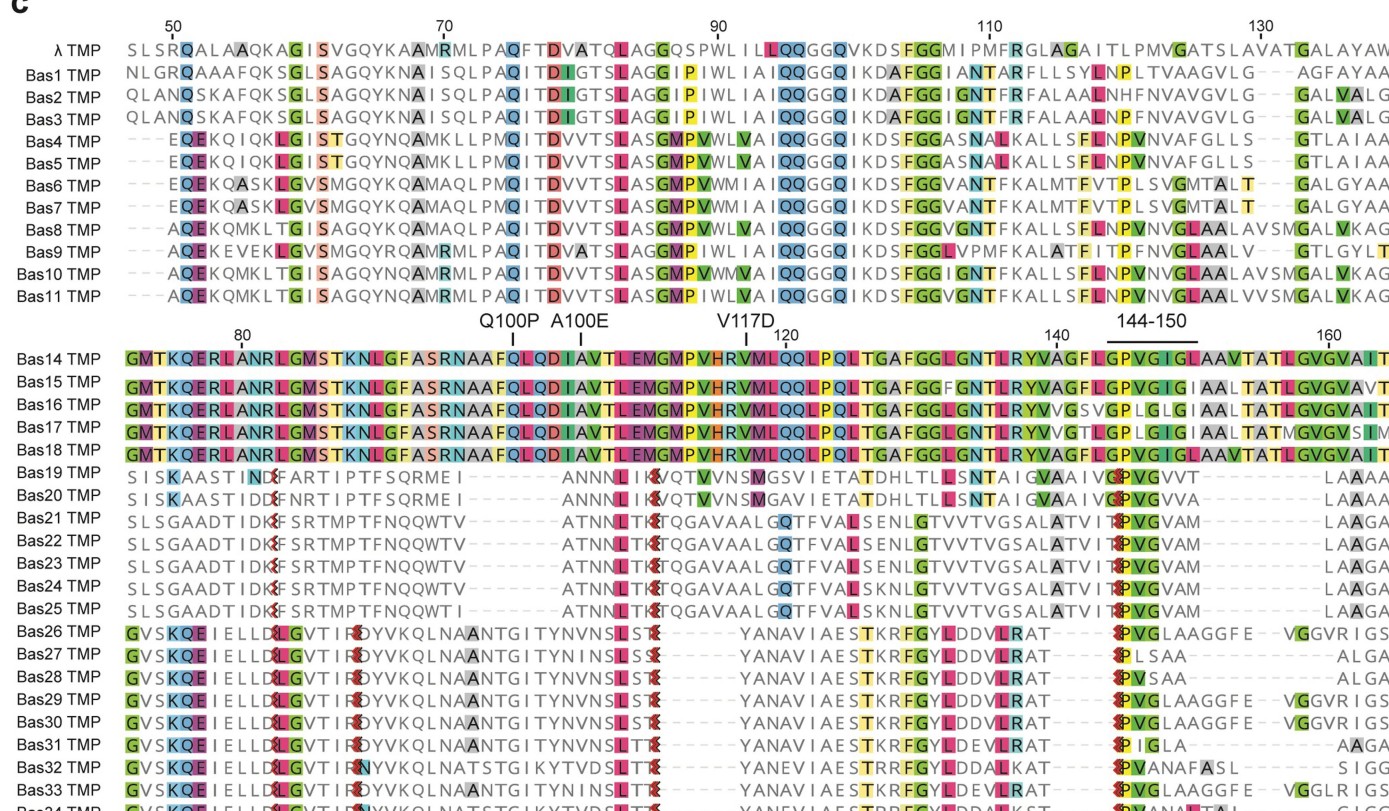

**Extended Data Fig. 7 | Bas14 escape phenotypes and TMP alignment. (a)** EOP data for Bas14 and Bas14 escape mutants on SNIPE and the indicated SNIPE mutants. **(b)** EOP data for Bas14 and Bas14 escape mutants on SNIPE. EOP was calculated by mixing phage and cells in top agar and using the top agar overlay method. Analysis was performed on n = 3 independent replicates. ** indicates p = 0.004 (Q100P), p = 0.003 (dup.144-150), p = 0.004 (A105E), p = 0.005 (Δ114-150), and p = 0.004 (V117D) (unpaired two-tailed t-tests). **(c)** Alignment of the region of interest from λ and BASEL siphovirus TMPs. Locations of Bas14 escape mutations are shown above the Bas14 TMP sequence. Amino acid numbers in the λ TMP and Bas14 TMP are shown for reference. Amino acids highlighted in color indicate positions identical to the Bas14 TMP reference sequence, with specific colors corresponding to different residues. Red chevrons indicate gaps that were hidden.

**a**

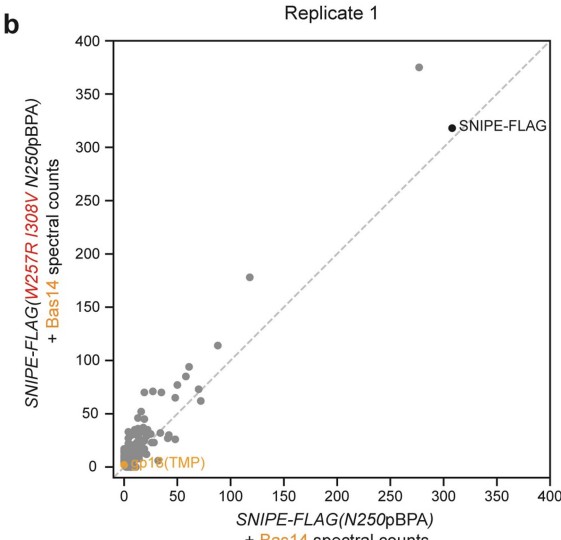
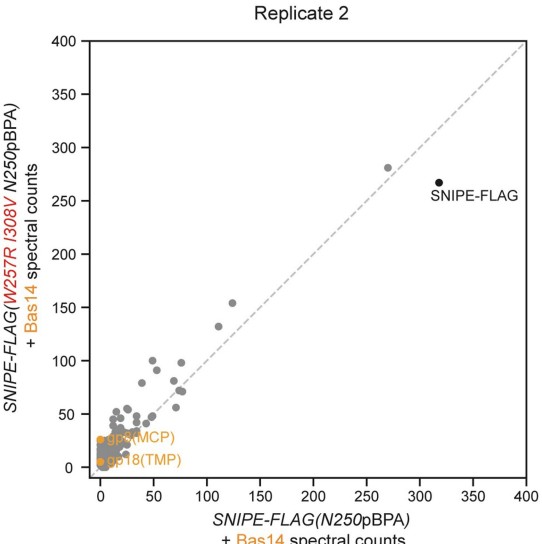

**b**

**Extended Data Fig. 8 | pBPA crosslinking during phage infection. (a)** Bas14 phage spotting on strains expressing a specialized tRNA sythetase (pBpaRS) and an empty vector or different SNIPE-FLAG constructs with 1.2 mM pBPA. (**b**) UV-induced crosslinking of Bas14-infected cells expressing pBpaRS and SNIPE-FLAG(W257R I308V N250pBPA) or SNIPE-FLAG(N250pBPA) in the presence of 1.2 mM pBPA. These proteins and crosslinked products were pulled down with anti-FLAG beads and quantified by mass spectrometry. Phage proteins are labeled in orange, and SNIPE-FLAG is labeled in black. n = 2 independent biological replicates are shown. Both replicates are identical to Fig. 4e but zoomed out to permit visualization of all data points.

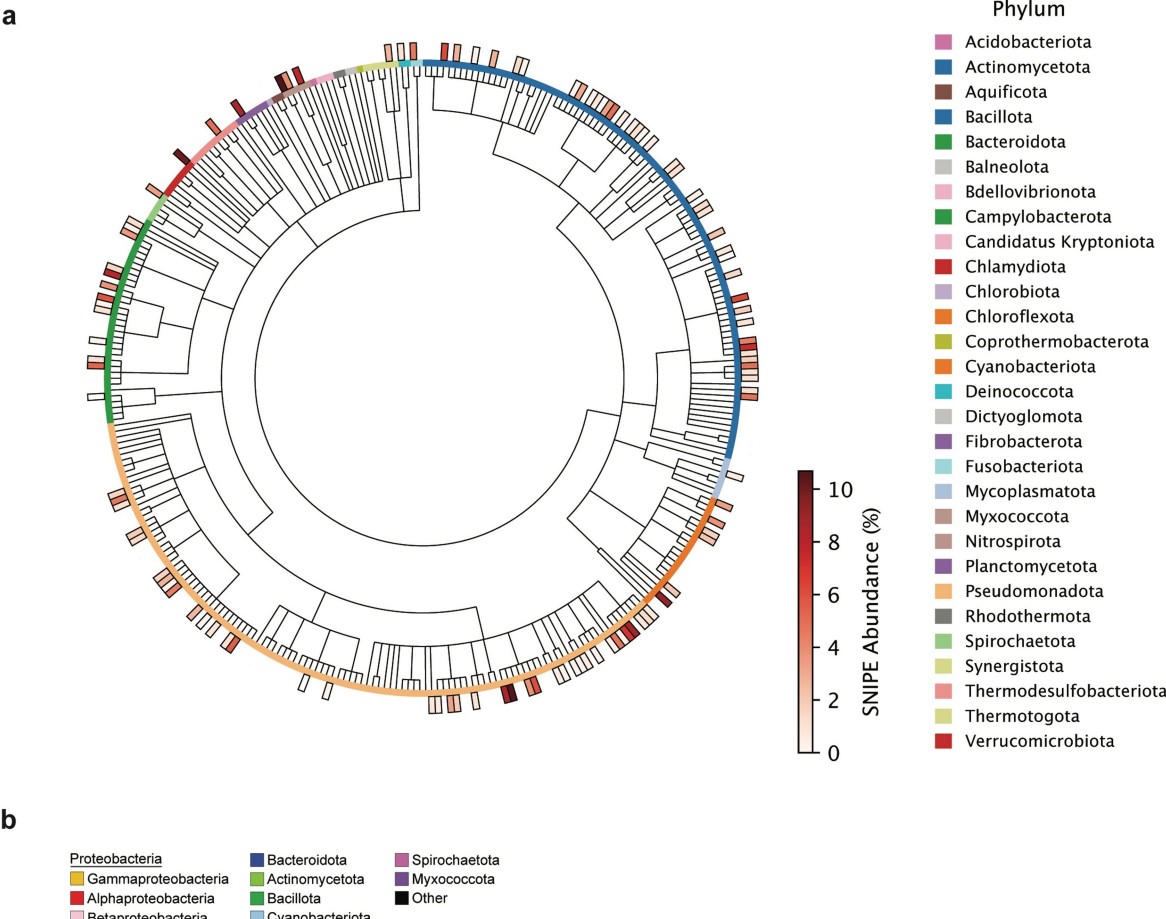

**a**

**Phylum**
- Acidobacteriota
- Actinomycetota
- Aquificota
- Bacillota
- Bacteroidota
- Balneolota
- Bdellovibrionota
- Campylobacterota
- Candidatus Kryptoniota
- Chlamydiota
- Chlorobiota
- Chloroflexota
- Coprothermobacterota
- Cyanobacteriota
- Deinococcota
- Dictyoglomota
- Fibrobacterota
- Fusobacteriota
- Mycoplasmatota
- Myxococcota
- Nitrospirota
- Planctomycetota
- Pseudomonadota
- Rhodothermota
- Spirochaetota
- Synergistota
- Thermodesulfobacteriota
- Thermotogota
- Verrucomicrobiota

SNIPE Abundance (%) — 0, 2, 4, 6, 8, 10

**b**

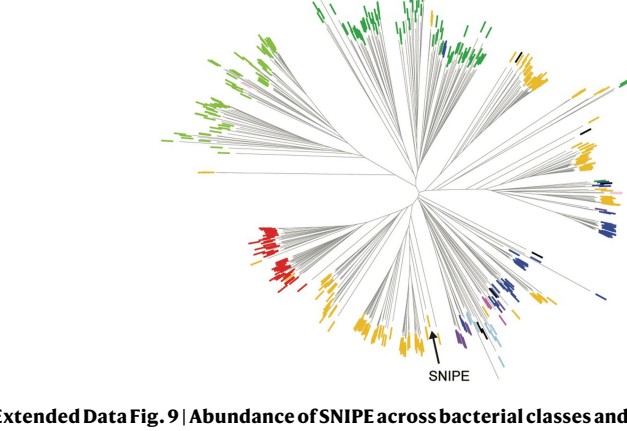

Proteobacteria
- Gammaproteobacteria
- Alphaproteobacteria
- Betaproteobacteria
- Bacteroidota
- Actinomycetota
- Bacillota
- Cyanobacteriota
- Spirochaetota
- Myxococcota
- Other

SNIPE

**Extended Data Fig. 9 | Abundance of SNIPE across bacterial classes and phyla. (a)** Rooted phylogenetic tree of bacterial families, with phyla indicated by color in the inner ring. The outer ring indicates the percentage of genomes within each family that contain at least one SNIPE homolog. Black outlines indicate families that harbor at least one genome containing a SNIPE homolog.

SNIPE homologs are as identified from Genbank and RefSeq genomes in a previous study[13]. Families are only shown if they contain ten or more genomes in the queried dataset. **(b)** Unrooted phylogenetic tree of 474 SNIPE homologs with tip colors representing bacterial classes (alpha-, beta-, and gamma-proteobacteria) or phyla (all other categories) that harbor these homologs.

**a**

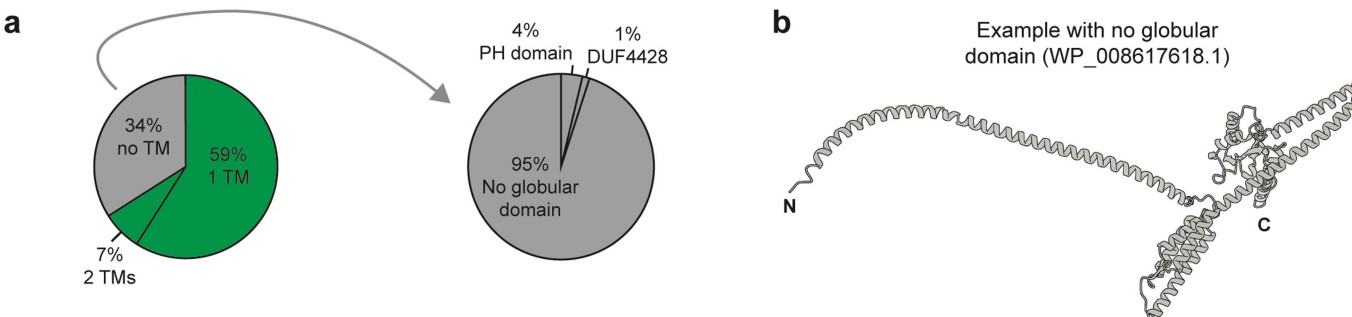

**b**

Example with no globular domain (WP_008617618.1)

**c**

Example with PH domain
(KEA32402.1)

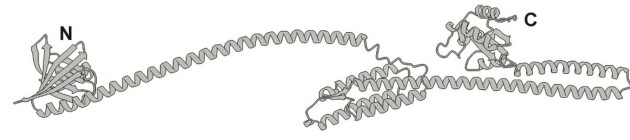

Top HHpred hit: PF14470.9 (bacterial PH domain) (p = 2 x 10^-8)
Top FoldSeek hit: GSK2 PH domain (PDB 2BCJ-A)

**d**

Example with DUF4428 domain
(WP_122023048.1)

Top HHpred hit: PF14471.9 (DUF4428) (p = 1 x 10^-15)

**e**

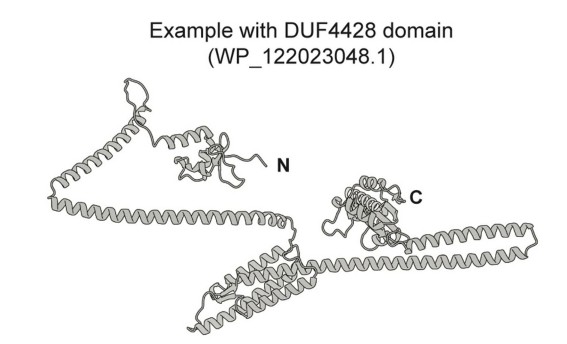

G-protein-coupled
Receptor Kinase 2 (GSK2)

SNIPE homolog
KEA32402.1
N-terminus

RMSD: 6.11

**f**

Putative DNA-binding
interface of SNIPE

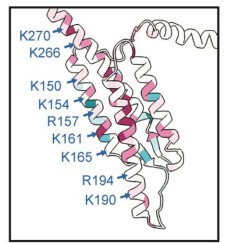

| Position | Residue variety |
|----------|-----------------|
| K150 | K 58%, G 9%, R 7% |
| K154 | K 27%, A 15%, D 14% |
| R157 | R 66%, K 16%, I 5% |
| K161 | K 87%, R 9%, A 1% |
| K165 | K 55%, R 21%, S 11% |
| K190 | K 50%, A 12%, T 11% |
| R194 | D 28%, R 23%, N 8% |
| K266 | K 62%, R 10%, L 9% |
| K270 | K 70%, R 19%, Q 4% |

Putative TMP-binding
interface of SNIPE

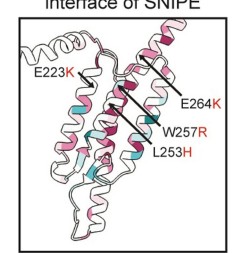

| Position | Residue variety |
|----------|-----------------|
| E223 | E 44%, R 11%, K 10% |
| L253 | L 48%, I 20%, V 15% |
| W257 | Y 20%, R 16%, K 12% |
| E264 | E 30%, L 19%, Q 14% |

**Extended Data Fig. 10 | Domains in N-terminal regions of SNIPE homologs.**
(**a**) Structures of the 34% of SNIPE homologs that lack a predicted TM were predicted by AlphaFold2 and the percentages of homologs with a globular domain at their N-terminus (PH or DUF4428) are shown. Homology was detected by FoldSeek and/or HHpred. (**b**) Example of a predicted SNIPE homolog structure generated by AlphaFold2. The structure lacks both a predicted transmembrane (TM) domain and a globular domain in the N-terminal region. (**c**) Example of a predicted SNIPE homolog structure generated by AlphaFold2. The structure lacks a predicted transmembrane (TM) domain and features a predicted pleckstrin homology (PH) domain at the N-terminus. Top matches from FoldSeek (using the PDB100 database of experimentally determined structures) and an HHpred search with the

N-terminal globular domain are presented. (**d**) Example of a predicted SNIPE homolog structure generated by AlphaFold2. The structure lacks a predicted transmembrane (TM) domain and features a predicted DUF4428 domain at the N-terminus. The top match from an HHpred search with the N-terminal globular domain is presented. (**e**) Structural alignment between the predicted PH domain of SNIPE homolog KEA32402.1 and the PH domain of GSK2 (PDB 2BCJ-A). (**f**) Same insets as shown in Fig. 5f, but with the percentage of different residues found at a given location across SNIPE homologs, as quantified by ConSurf. Only the top three most abundant residues are shown for each location. Positively charged amino acids are labeled in blue for the top inset. Mutations that enhanced defense against Bas14 are marked in red for the bottom inset.

# Reporting Summary

## Statistics

For all statistical analyses, confirm that the following items are present in the figure legend, table legend, main text, or Methods section.

| n/a | Confirmed | |
|---|---|---|
| ☐ | ☒ | The exact sample size (*n*) for each experimental group/condition, given as a discrete number and unit of measurement |
| ☐ | ☒ | A statement on whether measurements were taken from distinct samples or whether the same sample was measured repeatedly |
| ☐ | ☒ | The statistical test(s) used AND whether they are one- or two-sided<br>*Only common tests should be described solely by name; describe more complex techniques in the Methods section.* |
| ☒ | ☐ | A description of all covariates tested |
| ☒ | ☐ | A description of any assumptions or corrections, such as tests of normality and adjustment for multiple comparisons |
| ☐ | ☒ | A full description of the statistical parameters including central tendency (e.g. means) or other basic estimates (e.g. regression coefficient) AND variation (e.g. standard deviation) or associated estimates of uncertainty (e.g. confidence intervals) |
| ☐ | ☒ | For null hypothesis testing, the test statistic (e.g. *F*, *t*, *r*) with confidence intervals, effect sizes, degrees of freedom and *P* value noted<br>*Give P values as exact values whenever suitable.* |
| ☒ | ☐ | For Bayesian analysis, information on the choice of priors and Markov chain Monte Carlo settings |
| ☒ | ☐ | For hierarchical and complex designs, identification of the appropriate level for tests and full reporting of outcomes |
| ☐ | ☒ | Estimates of effect sizes (e.g. Cohen's *d*, Pearson's *r*), indicating how they were calculated |

*Our web collection on statistics for biologists contains articles on many of the points above.*

## Software and code

Policy information about availability of computer code

| Data collection | Consurf 2016 web server, FoldSeek Release 10, HHsuite v3.3.0 |
|---|---|
| Data analysis | MMseqs2 Release 14, UCSF ChimeraX v1.7, pandas v2.0.3, numpy v1.24.4, matplotlib v3.2.2, seaborn v0.10.1, Geneious version 2025.3, Fiji (ImageJ) version 2.1.0: watershed algorithm and coloc 2 plugin, AlphaFold version 3 web server |

For manuscripts utilizing custom algorithms or software that are central to the research but not yet described in published literature, software must be made available to editors and reviewers. We strongly encourage code deposition in a community repository (e.g. GitHub). See the Nature Portfolio guidelines for submitting code & software for further information.

## Data

Policy information about availability of data

All manuscripts must include a data availability statement. This statement should provide the following information, where applicable:
- Accession codes, unique identifiers, or web links for publicly available datasets
- A description of any restrictions on data availability
- For clinical datasets or third party data, please ensure that the statement adheres to our policy

Sequencing data is available in the Sequence Read Archive under BioProject PRJNA1231458. Summaries of spectral read counts and raw data for MS/MS of biotinylated or crosslinked proteins were deposited under MassIVE and can be accessed under accession MSV000097285. All other data are available in the manuscript or supplementary materials. Source data are provided for Figures 1-5 and Extended Data Figures 1, 3, 4, 6-10.

# Research involving human participants, their data, or biological material

Policy information about studies with human participants or human data. See also policy information about sex, gender (identity/presentation), and sexual orientation and race, ethnicity and racism.

| | |
|---|---|
| Reporting on sex and gender | N/A |
| Reporting on race, ethnicity, or other socially relevant groupings | N/A |
| Population characteristics | N/A |
| Recruitment | N/A |
| Ethics oversight | N/A |

Note that full information on the approval of the study protocol must also be provided in the manuscript.

# Field-specific reporting

Please select the one below that is the best fit for your research. If you are not sure, read the appropriate sections before making your selection.

☒ Life sciences    ☐ Behavioural & social sciences    ☐ Ecological, evolutionary & environmental sciences

For a reference copy of the document with all sections, see [nature.com/documents/nr-reporting-summary-flat.pdf](http://nature.com/documents/nr-reporting-summary-flat.pdf)

# Life sciences study design

All studies must disclose on these points even when the disclosure is negative.

| | |
|---|---|
| Sample size | No formal sample size calculation was performed. Sample sizes were selected based on prior experience and expected effect sizes, aiming to detect biologically meaningful differences while demonstrating reproducibility. Each experiment was independently replicated 2–4 times, which we determined is sufficient given the consistently large effect sizes observed. |
| Data exclusions | No data were excluded. |
| Replication | All experiments, except those noted below, were independently repeated at least twice, and all attempts were successful. TurboID with mass spectrometry experiments were only performed once due to cost and significant overlap across the reported experiments. Tn-Seq was only performed once due to recovery of expected hits such as ManY and ManZ with phage λ. |
| Randomization | All experiments were performed in isogenic strains so there were no covariates to control for. No subjective choice of experimental and control groups was performed. |
| Blinding | Blinding was not considered necessary because all data were objective and quantitative, such as sequencing reads, mass spectrometry intensities, or discrete counts. Raw data are reported in full in the manuscript, and all experiments were independently replicated with consistent results, minimizing the potential for bias in data collection or analysis. |

# Reporting for specific materials, systems and methods

We require information from authors about some types of materials, experimental systems and methods used in many studies. Here, indicate whether each material, system or method listed is relevant to your study. If you are not sure if a list item applies to your research, read the appropriate section before selecting a response.

## Materials & experimental systems

| n/a | Involved in the study |
|---|---|
| ☐ | ☒ Antibodies |
| ☒ | ☐ Eukaryotic cell lines |
| ☒ | ☐ Palaeontology and archaeology |
| ☒ | ☐ Animals and other organisms |
| ☒ | ☐ Clinical data |
| ☒ | ☐ Dual use research of concern |
| ☒ | ☐ Plants |

## Methods

| n/a | Involved in the study |
|---|---|
| ☒ | ☐ ChIP-seq |
| ☒ | ☐ Flow cytometry |
| ☒ | ☐ MRI-based neuroimaging |

## Antibodies

| Antibodies used | GFP Monoclonal (3E6) Mouse mAb, Invitrogen, Cat#: A11120<br>E. coli DnaK (1-384 aa) Rabbit pAb, AssayPro, Cat#: 32857-05111<br>OmpC Rabbit pAb, Bioss Antibodies, Cat#: bs20213R<br>Goat anti-Mouse IgG (H+L) Secondary Antibody, HRP, Thermo Fisher, Cat #: 32430<br>Goat anti-Rabbit IgG (H+L) Secondary Antibody, HRP, Thermo Fisher, Cat #: 32460 |
|---|---|
| Validation | Primary antibodies were validated both by the manufacturer and in our experiments. Manufacturer validation statements (from the supplier websites) are as follows:<br>anti-GFP: "Antibody specificity was demonstrated by detection of different targets fused to GFP tag in transiently transfected lysates tested."<br>anti DnaK: "Assay was performed using increasing concentrations of biotinylated recombinant E. coli DnaK protein"<br>anti-OmpC: "E. coli lysates probed with OmpC Polyclonal Antibody"<br>anti-Mouse: "Western blot analysis of HA Epitope Tag performed by various amounts of E. coli lysate containing a multi-epitope tagged protein"<br>anti-Rabbit: "Western blot analysis performed on membrane enriched extracts of K562 and PC-3."<br>In addition, we confirmed that each antibody produced a single band at the expected molecular weight in our samples and detected proteins in the appropriate cellular fraction (cytoplasmic or membrane), consistent with prior literature. Negative controls lacking the target protein produced no signal, supporting specificity. |

## Plants

| Seed stocks | Report on the source of all seed stocks or other plant material used. If applicable, state the seed stock centre and catalogue number. If plant specimens were collected from the field, describe the collection location, date and sampling procedures. |
|---|---|
| Novel plant genotypes | Describe the methods by which all novel plant genotypes were produced. This includes those generated by transgenic approaches, gene editing, chemical/radiation-based mutagenesis and hybridization. For transgenic lines, describe the transformation method, the number of independent lines analyzed and the generation upon which experiments were performed. For gene-edited lines, describe the editor used, the endogenous sequence targeted for editing, the targeting guide RNA sequence (if applicable) and how the editor was applied. |
| Authentication | Describe any authentication procedures for each seed stock used or novel genotype generated. Describe any experiments used to assess the effect of a mutation and, where applicable, how potential secondary effects (e.g. second site T-DNA insertions, mosiacism, off-target gene editing) were examined. |

