## [Peer Review file · Nature]

A membrane-bound nuclease directly cleaves phage DNA during genome injection

Corresponding Author: Professor Michael Laub

Version 0:

Reviewer comments:

Referee #1

(Remarks to the Author)

This study describes the mode of action of SNIPE (formerly known as PD- λ -1), a recently identified bacterial defense system that protects *E. coli* from phage λ infection. The authors conducted several cleverly designed experiments to demonstrate that SNIPE is a membrane-bound nuclease that cleaves phage λ DNA during genome injection. They further showed that SNIPE distinguishes λ DNA from host DNA by interacting with both the host mannose permease complex and the phage tape measure protein. In addition, SNIPE provides protection against other siphoviruses, although in these cases it operates independently of ManYZ. Taken together, this study highlights a novel, spatially organized strategy for viral recognition in bacteria.

Major Comments

1. Figure 1a: It is surprising that no growth curves (with and without phages as well as with and without SNIPE) are presented. Including these would likely support the claim that “these results confirm that SNIPE provides direct defense, enabling cells to ward off infection without notably compromising cell growth.”
2. Considering that SNIPE and ManYZ may interact *in vivo* prior to (and during) λ infection, does this affect the function of the mannose operon? Perhaps growth curves with various sugars could be performed.

Minor Comments

1. A line numbering system would have been appreciated.
2. Introduction: Please provide a reference for the statement: “In bacteria, various ‘direct defense’ systems also specifically target foreign DNA. This includes CRISPR-Cas systems and Argonautes, which use guide RNAs or DNAs, respectively, to cut foreign DNA in a sequence-specific manner.”
3. Figure 1b: Indicate the position of E414 within the GIY-YIG nuclease domain. Also, specify the overall size of the protein and the start and end points of its domains.
4. End of page 3: At this point, you cannot state “this toxicity was dependent on the nuclease activity of SNIPE(Δ TM), indicating that SNIPE(Δ TM) cuts host DNA when not membrane bound,” as no supporting evidence has yet been presented. You should also mention here that the Δ TM- Δ DUF4041 construct is detrimental to the cell (even though this is noted later on page 4).
5. Could the quality of Figures 1e and 1f be improved (e.g., colored)? They are currently difficult to evaluate properly, I had to zoom in significantly to discern the features the authors refer to.
6. Page 4: “These results indicated that the toxicity of SNIPE(Δ TM) stems from it localizing to and cleaving the host genome.” Since no direct evidence of host DNA cleavage is provided at this point, consider replacing “indicated” with “suggested.”
7. Extended Figure 2c: Gam-EcoRI appears to show more signals than Gam- Δ TM. Given that the Δ TM- Δ DUF4041 construct is also somewhat lethal, were you surprised to observe relatively few double-strand breaks?
8. Page 4: “Additionally, removal of the DUF4041 substantially reduced the toxicity of catalytically active SNIPE(Δ TM) (Fig. 1d).” This effect is observed at 40 μ M vanillate but not at 80 μ M. This discrepancy should be discussed.
9. Page 5: “In rare cases where a λ parS genome appeared in a SNIPE-containing cell, it went on to replicate and lyse the cell, suggesting that if phages stochastically evade SNIPE at the cell membrane, their development proceeds unimpeded.”

Could this outcome be influenced by the multiplicity of infection (MOI)?

10. Page 5: "These findings support the model that SNIPE cleaves phage DNA during, but not after, genome injection." If so, would you expect SNIPE to have no effect on plasmid transformation as well? Could this be tested?
11. Figure 4a: What does the "S" denote? Additionally, the MTP clades should be presented as they are in Figure 3f for consistency.
12. Page 7: "However, the large majority of siphoviruses were weakly targeted by SNIPE, with a subset targeted more robustly." It appears that about half of them were more robustly targeted.
13. Page 7: Do the siphoviruses from the Basel collection use Lamb as a receptor?
14. Page 8: Did you attempt to isolate lambda escapers? What about phages Bas15 to Bas18?
15. Page 8: For the phage escapers, were mutations found exclusively in the tape measure protein (TMP)?
16. Page 8: When comparing the TMP of phages Bas14 and Bas18 to those of Bas15–Bas17, do you have any hypotheses as to why they likely interact with different membrane proteins?
17. Page 8: Do phages Bas15–Bas17 use FhuA as a receptor?
18. Page 9: SNIPE homologs contain several diversified domains. While this is intriguing, I am curious about the natural diversity of SNIPE within *E. coli* strains. Is there any evidence of diversity that might suggest evolutionary adaptation of SNIPE proteins to changing TMPs?
19. Legend of Figure 6: Remove the second "The" at the beginning of the sentence.
20. References: The formatting of titles is inconsistent. Additionally, bacterial names should be italicized.

Referee #2

(Remarks to the Author)

In the manuscript by Saxton et al., the authors set out to characterize a new phage defense system, termed SNIPE, that provided defense against phage λ through a mechanism other than abortive infection. The authors elegantly demonstrate that SNIPE is targeted to the inner membrane where it is poised to cleave incoming phage DNA. SNIPE is composed of three domains: a transmembrane (TM) domain, a DUF4041, and a C-terminal nuclease domain. A combination of microscopy and cell fractionation were used to demonstrate the TM domain within SNIPE is required for membrane localization and in the absence of the TM domain SNIPE becomes toxic as it degrades the host DNA using the nuclease domain. Mutations suggest that DUF4041 enables association with DNA. Using radiolabeled phage genomes and microscopy the authors clearly show that newly injected phage DNA is cleaved by SNIPE and that SNIPE cannot target lysogens.

To better understand how SNIPE is targeted to infected phage DNA the authors investigated the proteins in close proximity to SNIPE. Their work suggests that SNIPE associates with the ManYZ. This was exciting as phage λ requires ManYZ to inject its genome into host cells. In further support of their hypothesis that SNIPE associates with ManYZ to localize to phage DNA, the authors found that SNIPE is able to defend against phages that use ManYZ for entry but not those that use other host proteins. Analysis of more distantly related phages demonstrated that mutant alleles of SNIPE can provide defense against phages that do not require ManYZ.

SNIPE and SNIPE-like systems can be found throughout bacteria suggesting this a widespread defense mechanism. While the nuclease domain is conserved, the authors found that the N-terminus is variable in composition and length but typically resembles inner membrane, or membrane associated proteins. The authors therefore put forward a model where the N-terminus is responsible for properly localizing SNIPE to where phage DNA is injected in the membrane, thereby preventing targeting of host DNA. Upon phage infection, DUF4041 binds to the phage tape measure protein and guides the nuclease to cleave the injected phage DNA thereby providing defense.

I complement the authors on an elegant story and creative approaches to tackling a challenging system. The majority of the claims are strongly supported, there are only a few places that require additional evidence to fully support the authors claims.

Major comments:

1. There is considerable discussion of discrimination between self and non-self DNA and the manuscript introduces the model that SNIPE sequestration at the membrane limits destruction of the host chromosome (autoimmunity). However, there is an ambiguous interaction with phage tape measure protein (TMP) that also plays a role in SNIPE function and an alternative hypothesis is that a phage protein is still required for full-length SNIPE activation. Can the authors show the relative contributions of subcellular localization and SNIPE activation when avoiding autoimmunity? A scenario that needs to be considered is that the Δ TM mutant may be both no longer sequestered at the membrane AND de-repressed by removal of autoinhibition facilitated by the TM domain. An ideal experiment would be to re-localize part of the chromosome to the membrane using an adaptation of the ParSB system by adding a TM domain (maybe even from ManYZ?) to ParB, then measuring growth +/- SNIPE. Is localization of the chromosome to the membrane sufficient for destruction? If no, one might expect that TMP is still a required molecular pattern for activating the SNIPE nuclease. In that case, further supplementation with TMP expressed from a plasmid should result in host chromosome destruction. This distinction is crucial because many chromosomal processes, including transcription, can happen at the membrane.

2. The relative roles for SNIPE interacting with inner membrane proteins such as ManYZ vs. TMP is confusing. The support for SNIPE interacting with ManYZ is based on proximity labeling and the λ generalist mutant. The support for SNIPE interacting with TMP are allelic suppressor mutations in BAS14 TMP. A unifying hypothesis could be that SNIPE mutations in Fig. 4b strengthen interactions with the inner membrane component that BAS14 requires for genome injection and that BAS14 escaper mutants in TMP weaken the interaction with the hypothetical inner membrane component. At the core of this

question is, does TMP interact with SNIPE? Can the authors either demonstrate a biochemical interaction between TMP and the SNIPE soluble domain OR show with proximity labeling that BAS14 TMP association with SNIPE is influenced by SNIPE mutations and/or BAS14 TMP mutations during BAS14 infection?

Minor comments:

1. The presentation of phage data as images or heat maps obscures much of the data as error and spread cannot be visualized. Can the data be graphed with individual points displayed?
2. The microscopy images may be difficult to interpret in their current form (i.e. Fig. 1e). Could zoomed-in inset images be shown for individual cells? False coloring and overlays could also be used to support colocalization of GFP and DAPI or the membrane.
3. Additional controls and quantification of some of the microscopy would also help support the authors claims. For example, including a membrane dye within Fig.1 would support that SNIPE localizes to the membrane. Additionally, calculating the Pearson's correlation coefficient for DAP1 and GFP would support that SNIPE localized to host DNA in the absence of the TM domain.
4. Please define how EOP was calculated in the figure legends and/or methods section.
5. For Fig. 4a, the 's' symbol within certain squares is not defined.
6. Figure 5 presentation could be improved to highlight the modularity of SNIPE. The use of the pie chart obscures/underwhelms many of the findings. Could representative examples of SNIPE with the unique domain architecture be shown using cartoon depictions? Or perhaps demonstrating the structural similarity between the N-term and some of the hits listed in the pie chart? Especially the non-TM group would be interesting.
7. Within the section entitled "SNIPE interacts with the mannose permease complex" the line "To our knowledge, this is the first documented example of a phage protein interacting with a bacterial inner membrane protein during genome infection" is unclear and unsupported. I think the authors mean "...interacting with a an antiphage inner membrane protein...". Also, further work is required to verify this SNIPE directly interacts with ManYZ.
8. Figure 3c is misleading. The authors are comparing protein abundance plus and minus phage infection, however, because they are measuring abundance of a phage protein and there is only phage protein in one sample, the graph artificially illustrates an enrichment of TMP. The more accurate comparison is in Fig. 3e and 3c should be removed or placed in the supplement.
9. Please provide western blot analysis of the SNIPE mutants used throughout (Δ DUF4041, Δ TM, etc.) to demonstrate that protein expression levels are not affected. These data are crucial to validate the authors conclusions.

Version 1:

Reviewer comments:

Referee #1

(Remarks to the Author)

I appreciate that the authors have added new information. I have a few additional comments (see below).

1- Regarding the interaction of SNIPE with the mannose operon, why growth curves with mannose were not performed? I would have expected growth curves in minimal medium with mannose at the sole carbon source (and with glucose as a control). The MacConkey agar assay, indicate that the mannose operon is still functional but I was wondering of its efficacy might be impair.

2- MOI and spherical cells, at which MOIs did you observed this phenomenon? It is unclear why the plasmid transformation data is not added in supp mat?

3- The information that you attempted but failed to isolate λ escapers by plating for single plaques and also performing liquid evolution experiments should be in the manuscript as data not shown.

Referee #2

(Remarks to the Author)

The authors have satisfied all of my critiques and I congratulate them on their high-quality work.

We thank the reviewers for their positive and constructive feedback. Below we respond to each query and indicate, when appropriate, how the text and/or figures have been updated.

Referees' comments:

Referee #1 (Remarks to the Author):

This study describes the mode of action of SNIPE (formerly known as PD- λ -1), a recently identified bacterial defense system that protects *E. coli* from phage λ infection. The authors conducted several cleverly designed experiments to demonstrate that SNIPE is a membrane-bound nuclease that cleaves phage λ DNA during genome injection. They further showed that SNIPE distinguishes λ DNA from host DNA by interacting with both the host mannose permease complex and the phage tape measure protein. In addition, SNIPE provides protection against other siphoviruses, although in these cases it operates independently of ManYZ. Taken together, this study highlights a novel, spatially organized strategy for viral recognition in bacteria.

Major Comments

1. Figure 1a: It is surprising that no growth curves (with and without phages as well as with and without SNIPE) are presented. Including these would likely support the claim that “these results confirm that SNIPE provides direct defense, enabling cells to ward off infection without notably compromising cell growth.”

We thank the reviewer for this helpful suggestion. We have now included growth curves (Figure 1b; lines 77-78), which provide stronger support for our conclusion that cells harboring SNIPE maintain normal growth during phage infection.

2. Considering that SNIPE and ManYZ may interact in vivo prior to (and during) λ infection, does this affect the function of the mannose operon? Perhaps growth curves with various sugars could be performed.

We thank the reviewer for raising this important point. To assess whether SNIPE binding affects the normal function of ManYZ, we performed a MacConkey agar assay, a well-established approach in which mannose metabolism acidifies cells and shifts the color of a pH indicator in the medium. The results (Extended Data Fig. 4h; lines 248-250) indicate that SNIPE does not impair the ability of ManYZ to transport or metabolize mannose.

Minor Comments

1. A line numbering system would have been appreciated.

We apologize for this oversight and have now included page numbers.

2. Introduction: Please provide a reference for the statement: “In bacteria, various ‘direct defense’ systems also specifically target foreign DNA. This includes CRISPR-Cas systems and Argonautes, which use guide RNAs or DNAs, respectively, to cut foreign DNA in a sequence-specific manner.”

We thank the reviewer for catching this omission. We have now added two references (one for Argonautes and one for CRISPR-Cas) in the Introduction (lines 51-53).

3. Figure 1b: Indicate the position of E414 within the GIY-YIG nuclease domain. Also, specify the overall size of the protein and the start and end points of its domains.

What was previously Figure 1b is now Figure 1c, and has a more extensive schematic detailing the start and end points of SNIPE domains, overall protein length, and the location of the E414A mutation.

4. End of page 3: At this point, you cannot state “this toxicity was dependent on the nuclease activity of SNIPE(Δ TM), indicating that SNIPE(Δ TM) cuts host DNA when not membrane bound,” as no supporting evidence has yet been presented. You should also mention here that the Δ TM- Δ DUF4041 construct is detrimental to the cell (even though this is noted later on page 4). We appreciate the reviewer’s comments and have revised the logic flow to more directly support our claims. For clarity, lines 101-107 now state:

"Attempts to clone SNIPE lacking the transmembrane domain (Δ TM) were unsuccessful, so we put SNIPE(Δ TM) under the control of a vanillate inducible (P_{van}) promoter. Induced expression of this construct was highly toxic, and this toxicity was abolished by the E414A mutation (the toxicity was also partially reduced by Δ DUF4041, which is explored below) (Fig. 1e). These data indicated that SNIPE(Δ TM) may localize to and cleave host DNA. To test this model, we used Gam-GFP, a fluorescently tagged RecBCD inhibitor that localizes to double strand DNA breaks in vivo..."

We believe this revised structure better connects the proposed model with the subsequent Gam-GFP experiment, and the role of DUF4041 is now discussed further in lines 138–150.

5. Could the quality of Figures 1e and 1f be improved (e.g., colored)? They are currently difficult to evaluate properly, I had to zoom in significantly to discern the features the authors refer to. We appreciate this idea and have updated the microscopy data to be more zoomed in for what is now Figures 1f-g, Extended Data Fig. 1f, Extended Data Fig. 2d and 2f. This aids interpretations of subcellular localization of SNIPE-GFP and other fluorescent constructs. We have also provided merged and colored images for these figure sections for better comparison across channels.

6. Page 4: “These results indicated that the toxicity of SNIPE(Δ TM) stems from it localizing to and cleaving the host genome.” Since no direct evidence of host DNA cleavage is provided at this point, consider replacing "indicated" with "suggested."

We have updated this, which can now be found in lines 105-106.

7. Extended Figure 2c: Gam-EcoRI appears to show more signals than Gam- Δ TM. Given that the Δ TM- Δ DUF4041 construct is also somewhat lethal, were you surprised to observe relatively few double-strand breaks?

This was also our interpretation, and we have now bolstered it with quantification of maximum Gam-GFP focus intensity per cell (now found in Extended Data Fig. 1f). We find this to be the best quantitative proxy for Gam-GFP foci given that these foci tend to coalesce, making enumeration difficult.

In response to the reviewer's question, we do observe fewer/less intense Gam-GFP foci in the $\Delta TM \Delta DUF4041$ condition as compared to ΔTM . We find these data consistent with our bacterial spotting data in Fig. 1e, which argue that deletion of the DUF4041 reduces but doesn't eliminate the toxicity observed with SNIPE(ΔTM).

8. Page 4: “Additionally, removal of the DUF4041 substantially reduced the toxicity of catalytically active SNIPE(ΔTM) (Fig. 1d).” This effect is observed at 40 μM vanillate but not at 80 μM . This discrepancy should be discussed.

We believe this is not a discrepancy and is consistent with our other data. In what is now Fig. 1g and Extended Data Fig. 1f, we show that the DUF4041 promotes binding to DNA, with the nuclease domain cleaving DNA. So, we suspect that at 40 μM vanillate, SNIPE(ΔTM) readily binds to and cleaves host DNA, whereas SNIPE($\Delta TM \Delta DUF4041$) has a low enough affinity for DNA that toxicity is not observed. In contrast, at 80 μM vanillate, SNIPE($\Delta TM \Delta DUF4041$) now likely interacts more readily with the chromosome, leading to greater toxicity (though the toxicity is still not as strong as SNIPE(ΔTM)).

9. Page 5: “In rare cases where a $\lambda parS$ genome appeared in a SNIPE-containing cell, it went on to replicate and lyse the cell, suggesting that if phages stochastically evade SNIPE at the cell membrane, their development proceeds unimpeded.” Could this outcome be influenced by the multiplicity of infection (MOI)?

We attempted to perform a variant of the experiment shown in Fig. 2b-c with a range of higher MOIs to test this question. However, we encountered an issue in which higher MOIs lead to a

“lysis-from-without” phenomenon in which cells immediately become spherical upon exposure to phage (see figure below). We believe this is due to (1) genome injection by several phages at once, and (2) a necessary step in our protocol in which we incubate cells and phage on ice for 30 minutes to facilitate adsorption while preventing genome injection, a convention in the λ field for synchronizing genome injection which may also inadvertently disrupt normal

membrane dynamics. Anecdotally, we have seen this lysis-from-without issue largely disappear in the absence of the incubation on ice, but we hesitate to use these data because of the consequent asynchronicity of genome entry. In the experiment shown here, we used the highest MOI possible without making all cells spontaneously turn spherical (though you can still see examples) to produce some data for consideration by reviewers. Our qualitative interpretation is that this higher MOI does not overwhelm SNIPE, but quantification was difficult due to challenges distinguishing foci when the number of foci was increased overall.

10. Page 5: “These findings support the model that SNIPE cleaves phage DNA during, but not after, genome injection.” If so, would you expect SNIPE to have no effect on plasmid transformation as well? Could this be tested?

We have performed this analysis (see figure to the right) and found that SNIPE does not affect plasmid transformation. These data are provided here for consideration by reviewers. However, we believe the message that SNIPE doesn't target all DNA crossing the membrane is also conveyed by its inability to target a large series of tested myoviruses and podoviruses, as well as our observations that relatively minor mutations in phage tail proteins can severely disrupt SNIPE-mediated defense. Therefore, we excluded this analysis from the paper.

11. Figure 4a: What does the "S" denote? Additionally, the MTP clades should be presented as they are in Figure 3f for consistency.

We apologize for the oversight in our initial submission regarding the missing explanation for "S." We have now updated the figure legends and corresponding figures (Fig. 4a and Extended Data Fig. 6c) to clarify that "S" denotes smaller plaques relative to those on the control bacterial strain. We have also revised the presentation of MTP clades to ensure consistent ordering between Fig. 3d and Extended Data Fig. 5a–b.

12. Page 7: "However, the large majority of siphoviruses were weakly targeted by SNIPE, with a subset targeted more robustly." It appears that about half of them were more robustly targeted. We apologize for the confusion caused by our lack of definition for "S" in the initial submission. "S" denotes smaller plaques relative to those on the control strain. With this clarification, we note that 24 of 32 siphoviruses (75%) in the BASEL collection are defended against by SNIPE, supporting our description of a "large majority" (Fig. 4a, lines 286-287).

13. Page 7: Do the siphoviruses from the Basel collection use LamB as a receptor? Bas20-24 use LamB, and other BASEL siphoviruses use FhuA, BtuB, LptD, TolC, and YncD (Maffei et. al 2021 *Plos Biology*).

14. Page 8: Did you attempt to isolate lambda escapers? What about phages Bas15 to Bas18? We attempted to isolate λ escapers by plating for single plaques and also performing liquid evolution experiments. Neither of these was successful. The only way we could get λ phages that escape SNIPE is in the $\Delta manYZ$ background, which yields mutants with identical phenotypes to the λ generalist mutant shown in Fig. 3a-b.

We did isolate escapers for Bas15 and Bas18. Bas15 escapers revert a residue in the tail completion protein gp13 (equivalent to gpZ in λ) to the wild-type residue that is present in the rest of the Bas14-18 clade (see below). Interestingly, gpZ was very weakly labeled by TurboID-ManZ in our proximity labeling experiments, has a proposed but enigmatic role in genome injection, and associates with the TMP in the phage virion structure (Thomas, Sternberg, & Weisberg 1978 *J. Mol. Biol.*, Kizziah et al. 2025, *Structure*).

In contrast, Bas18 escapers mutate an auxiliary gene (gp25) that is just downstream of the gene that encodes the tail spike (see below). The mutated gene has no known function, but its location next to the tail spike may indicate that it has some role in genome injection or superinfection exclusion.

Given that we have no clear models for how these mutations alter phage susceptibility to SNiPE-mediated defense, we believe that these data would be confusing for readers and therefore beyond the scope of this paper. However, it does speak to a complex interplay between SNiPE and phage genome injection machinery, and points to exciting avenues for future work.

15. Page 8: For the phage escapers, were mutations found exclusively in the tape measure protein (TMP)?

We performed whole genome sequencing and found that the only mutations in Bas14 escapers were in the tape measure protein. We have now updated lines 311-312 to reflect this, which is copied below.

"Strikingly, the only mutations found in escape phages mapped to the Bas14 gene that encodes the tape measure protein (Fig. 4d)."

16. Page 8: When comparing the TMP of phages Bas14 and Bas18 to those of Bas15–Bas17, do you have any hypotheses as to why they likely interact with different membrane proteins?

We thank the reviewer for this question, which prompted us to examine the Bas14–18 TMP clade more closely. We did not identify any regions of the Bas14 and Bas18 TMPs that differ substantially from those of Bas15–17. This prompted us to look into the literature, which argues that several phages that use FhuA as a receptor also require TonB for adsorption, given that these two proteins physically interact. (Braun et al. 2023, Journal of Bacteriology). This motivated us to test whether Bas15–17, which also use FhuA as a receptor, use TonB. Indeed, $\Delta tonB$ disrupted adsorption of Bas15 to the same degree of $\Delta fhuA$. Therefore, the difference in receptor specificity between Bas14/18 (which uses LptD, an essential host protein) and Bas15–17 (which uses FhuA), previously documented by Maffei et. al 2021 *Plos Biology*), explains the dependence of Bas15–17 on TonB.

As a result of this analysis, we have replaced the original Bas14–18 Tn-Seq data with a new experiment directly comparing infection by Bas14 to one of its escaper phages. This revised logic is now described in lines 323–340 and presented in Extended Data Fig. 8a.

"How do these SNiPE mutants provide enhanced defense in a manner that is dependent on the tape measure protein? One possibility is that SNiPE binds weakly to diverse siphovirus tape measure proteins, and the SNiPE mutants strengthen this interaction for phages in the Bas14-18 clade. Alternatively, the SNiPE mutants may enhance binding to

an inner membrane protein used by the Bas14-18 clade, and Bas14 escaper mutations may switch to using a different inner membrane protein. To test the latter possibility, we performed transposon-insertion sequencing (Tn-Seq)³¹. Specifically, we used λ , Bas14, or the Bas14 *TMP(A105E)* escape isolate to infect pools of cells in which barcoded transposons are inserted throughout the genome, such that most cells will die from infection, but cells harboring a transposon in a gene necessary for phage infection will survive. As expected, this screen identified ManY and ManZ as required for infection of λ (Extended Data Fig. 8a). In contrast, no integral inner membrane proteins were required for infection of either Bas14 or Bas14 *TMP(A105E)*. Notably, one or both of these phages may require essential inner membrane proteins, which cannot be identified with Tn-Seq. Nevertheless, we find it unlikely that relatively minor mutations in Bas14 escapers conferred a complete switch from one essential inner membrane protein to another, and instead favor a model in which these phages do not use a specific host inner membrane protein for genome injection. By extension, our results indicate that the SNIPE mutants may enhance binding to the Bas14 tape measure protein, which is disrupted in Bas14 escapers."

17. Page 8: Do phages Bas15–Bas17 use FhuA as a receptor?

Yes, this was demonstrated in (Maffei et. al 2021 *Plos Biology*) and was also visible in our previous Tn-Seq data. We also confirmed this with plaquing assays on $\Delta fhuA$.

18. Page 9: SNIPE homologs contain several diversified domains. While this is intriguing, I am curious about the natural diversity of SNIPE within E. coli strains. Is there any evidence of diversity that might suggest evolutionary adaptation of SNIPE proteins to changing TMPs?

We really appreciate this suggestion, as it prompted us to delve further into the ConSurf analysis performed in Figure 5g and Extended Data Fig. 10g. Our analysis showed that two mutations that enhanced SNIPE defense against Bas14 (E223K and W257R) are commonly found in diverse SNIPE homologs. This is especially noteworthy for W257R, given that R is far more common than W at the 257 position across SNIPE homologs and the W257R mutation gave the strongest enhanced defense against Bas14. We updated Extended Data Fig. 10g to include these analyses, and reference this in lines 411-413:

"In particular, the E223K and W257R mutations that were sufficient to enhance defense against Bas14-18 are frequently found in SNIPE homologs (Extended Data Fig. 10g)."

19. Legend of Figure 6: Remove the second "The" at the beginning of the sentence.

We thank the reviewer for catching this, it has been fixed.

20. References: The formatting of titles is inconsistent. Additionally, bacterial names should be italicized.

We appreciate this comment and have updated formatting of reference titles.

Referee #2 (Remarks to the Author):

In the manuscript by Saxton et al., the authors set out to characterize a new phage defense

system, termed SNIPE, that provided defense against phage λ through a mechanism other than abortive infection. The authors elegantly demonstrate that SNIPE is targeted to the inner membrane where it is poised to cleave incoming phage DNA. SNIPE is composed of three domains: a transmembrane (TM) domain, a DUF4041, and a C-terminal nuclease domain. A combination of microscopy and cell fractionation were used to demonstrate the TM domain within SNIPE is required for membrane localization and in the absence of the TM domain SNIPE becomes toxic as it degrades the host DNA using the nuclease domain. Mutations suggest that DUF4041 enables association with DNA. Using radiolabeled phage genomes and microscopy the authors clearly show that newly injected phage DNA is cleaved by SNIPE and that SNIPE cannot target lysogens.

To better understand how SNIPE is targeted to infected phage DNA the authors investigated the proteins in close proximity to SNIPE. Their work suggests that SNIPE associates with the ManYZ. This was exciting as phage λ requires ManYZ to inject its genome into host cells. In further support of their hypothesis that SNIPE associates with ManYZ to localize to phage DNA, the authors found that SNIPE is able to defend against phages that use ManYZ for entry but not those that use other host proteins. Analysis of more distantly related phages demonstrated that mutant alleles of SNIPE can provide defense against phages that do not require ManYZ. SNIPE and SNIPE-like systems can be found throughout bacteria suggesting this a widespread defense mechanism. While the nuclease domain is conserved, the authors found that the N-terminus is variable in composition and length but typically resembles inner membrane, or membrane associated proteins. The authors therefore put forward a model where the N-terminus is responsible for properly localizing SNIPE to where phage DNA is injected in the membrane, thereby preventing targeting of host DNA. Upon phage infection, DUF4041 binds to the phage tape measure protein and guides the nuclease to cleave the injected phage DNA thereby providing defense.

I complement the authors on an elegant story and creative approaches to tackling a challenging system. The majority of the claims are strongly supported, there are only a few places that require additional evidence to fully support the authors claims.

Major comments:

1. There is considerable discussion of discrimination between self and non-self DNA and the manuscript introduces the model that SNIPE sequestration at the membrane limits destruction of the host chromosome (autoimmunity). However, there is an ambiguous interaction with phage tape measure protein (TMP) that also plays a role in SNIPE function and an alternative hypothesis is that a phage protein is still required for full-length SNIPE activation. Can the authors show the relative contributions of subcellular localization and SNIPE activation when avoiding autoimmunity? A scenario that needs to be considered is that the Δ TM mutant may be both no longer sequestered at the membrane AND de-repressed by removal of autoinhibition facilitated by the TM domain. An ideal experiment would to re-localize part of the chromosome to the membrane using an adaptation of the ParSB system by adding a TM domain (maybe even from ManYZ?) to ParB, then measuring growth +/- SNIPE. Is localization of the chromosome to the membrane sufficient for destruction? If no, one might expect that TMP is still a required molecular pattern for activating the SNIPE nuclease. In that case, further supplementation with TMP expressed from a plasmid should result in host chromosome destruction. This distinction is

crucial because many chromosomal processes, including transcription, can happen at the membrane.

We agree that the ability (or inability) of SNIPE to cleave host DNA at the cell membrane was not fully addressed, and that deleting the transmembrane domain may both relocalize and activate the SNIPE nuclease domain. To this end, we first attempted to perform the reviewer's suggested experiment in which ParB could recruit DNA to the cell membrane.

Specifically, we inserted *parS* into the *E. coli* genome and fused the membrane protein LacY to CFP-ParB. We observed that this fusion protein localized to the cell membrane, but we no longer observed CFP-ParB foci and DAPI-stained DNA did not appear to localize to the cell membrane (data are shown on the right, top panel). Therefore, we decided that this approach was not feasible to test the stated hypothesis.

We reasoned that a less complex DNA-binding protein may circumvent these challenges and allow us to localize a subset of host DNA to the cell membrane. To this end, we fused two transmembrane domains from the membrane protein MalF to GFP-Fis; Fis is a nucleoid-associated protein in *E. coli*.

We observed that this fusion protein could efficiently localize DAPI-stained DNA to the cell membrane (bottom panel in the figure). This effect was toxic to cells, but we reasoned that if SNIPE can cleave host DNA at the cell membrane, cells expressing MalF(TM1-2)-GFP-Fis and SNIPE would show additive toxicity across a range of MalF(TM1-2)-GFP-Fis expression levels.

However, we did not observe any such added toxicity from SNIPE. These experiments are now described in lines 127-137 (shown below) and can be found in Extended Data Fig. 2d.

"Given that bacterial DNA likely contacts the cell membrane during cellular processes such as chromosome replication and segregation²¹, yet SNIPE is not intrinsically toxic to cells, we hypothesized that membrane-localized SNIPE does not cleave membrane-localized host DNA. To test this, we ectopically localized host DNA to the cell membrane by fusing two transmembrane domains from MalF to GFP-Fis, a fluorescently tagged DNA-binding protein. Strong expression of MalF(TM1-2)-GFP-Fis localized DAPI-stained DNA to the cell membrane and was highly toxic even in the absence of SNIPE (Extended Data Fig. 2d). Across a range of MalF(TM1-2)-GFP-Fis expression levels with varying degrees of toxicity, the presence of SNIPE generated no additional toxicity for cells. This result indicates that localization of host DNA to the cell membrane does not make it susceptible to SNIPE-mediated cleavage, suggesting that SNIPE exists in an auto-inhibited state when localized to the membrane."

Regarding the idea of testing whether ectopic production of TMP can lead to additional SNIPE activity: this is an excellent idea in principle, but TMP produced in the cell will not insert

properly into the inner membrane and form the same structure it adopts when the protein is ejected from the phage's tail tube during phage genome injection.

2. The relative roles for SNIPE interacting with inner membrane proteins such as ManYZ vs. TMP is confusing. The support for SNIPE interacting with ManYZ is based on proximity labeling and the λ generalist mutant. The support for SNIPE interacting with TMP are allelic suppressor mutations in BAS14 TMP. A unifying hypothesis could be that SNIPE mutations in Fig. 4b strengthen interactions with the inner membrane component that BAS14 requires for genome injection and that BAS14 escaper mutants in TMP weaken the interaction with the hypothetical inner membrane component. At the core of this question is, does TMP interact with SNIPE? Can the authors either demonstrate a biochemical interaction between TMP and the SNIPE soluble domain OR show with proximity labeling that BAS14 TMP association with SNIPE is influenced by SNIPE mutations and/or BAS14 TMP mutations during BAS14 infection?

We appreciate this astute observation and the suggestions. To address this, as mentioned above, we performed a new Tn-Seq experiment aimed at identifying inner membrane proteins used by Bas14 and one of the Bas14 escapers (*TMP A105E*). However, we were unable to identify inner membrane proteins used by either of these phages, which eliminated the possibility of using TurboID-inner membrane protein fusions (such as TurboID-ManZ in Figure 3) to measure SNIPE recruitment to Bas14 genome injection sites. Additionally, SNIPE-TurboID was non-functional and thus could not be used to identify its binding partners.

To provide an orthogonal method for measuring transient protein-protein interactions, we used unnatural amino acid crosslinking with *p*-benzoylphenylalanine. The logic for this experiment is now described in lines 341-358 (see below) and data can be found in Fig. 4e and Extended Data Fig. 8b-d.

"To test if the SNIPE DUF4041 physically interacts with the Bas14 tape measure protein, we turned to crosslinking with the unnatural amino acid *p*-benzoylphenylalanine (pBPA). In this approach, we replaced the SNIPE N250 codon with an amber codon and expressed a specialized tRNA synthetase that can incorporate pBPA at this site³². Given that N250 is adjacent to the W257R mutation that enhanced defense against Bas14, we reasoned pBPA incorporated at this site may crosslink to the Bas14 tape measure protein (Fig. 4c). First, we confirmed that SNIPE-FLAG(W257R I308V N250pBPA) was still able to defend against Bas14 (Extended Data Fig. 8b). Next, we infected this strain with Bas14, exposed cells to UV to crosslink pBPA to nearby proteins, and used anti-FLAG pulldowns to identify crosslinked proteins with mass spectrometry. The only phage protein recovered across both replicates was the tape measure protein (Fig. 4e, Extended Data Fig. 8c-d). In contrast, no phage proteins were detected under similar conditions with SNIPE-FLAG(N250pBPA), suggesting that recovery of phage proteins was dependent on the W257R I308V mutations. Together, these results indicate that the W257R I308V mutations enhance binding to the Bas14 tape measure protein. Therefore, our results support a model in which wild-type SNIPE provides broad defense against siphoviruses by weakly interacting with siphovirus tape measure proteins. This defense could then be augmented by SNIPE binding to an inner membrane protein used by the phage, like ManYZ as with λ , or by a strengthened interaction between the DUF4041 and

a specific type of tape measure protein, as with Bas14."

We note that the recovery of crosslinked phage proteins was quite low, which likely reflects the low efficiency of crosslinking and limited amount of phage protein substrate in the cell. However, the specific recovery of tape measure protein peptides in independent replicates suggests that the DUF4041 and the TMP physically interact. We do find this to be an unusually complex set of interactions that can dictate the targeting specificity of SNIPE (binding to inner membrane proteins and/or phage TMPs), but it provides the best explanation for the findings that SNIPE defense against λ largely relies on ManYZ, while SNIPE can also defend against other phages independent of ManYZ.

Minor comments:

1. The presentation of phage data as images or heat maps obscures much of the data as error and spread cannot be visualized. Can the data be graphed with individual points displayed?

We have updated bar graphs to include data points for different replicates (for example, Fig. 2a, c, f).

2. The microscopy images may be difficult to interpret in their current form (i.e. Fig. 1e). Could zoomed-in inset images be shown for individual cells? False coloring and overlays could also be used to support colocalization of GFP and DAPI or the membrane.

We thank the reviewer for this suggestion and have made all microscopy images focused on subcellular localization (i.e. Fig. 1g) more zoomed in. These images now also have merged, pseudocolored panels to support colocalization claims.

3. Additional controls and quantification of some of the microscopy would also help support the authors claims. For example, including a membrane dye within Fig.1 would support that SNIPE localizes to the membrane. Additionally, calculating the Pearson's correlation coefficient for DAPI and GFP would support that SNIPE localized to host DNA in the absence of the TM domain.

We calculated the Pearson's correlation coefficients for Fig. 1g, Extended Data Fig. 2d, and Extended Data Fig. 2f, which are now included in these figures. Otherwise, we believe that the combination of SNIPE-GFP localization, PhoA-LacZ experiments, the cytoplasm-membrane fractionation followed by immunoblotting offer multiple orthogonal tests suggesting that SNIPE is localized to the cell membrane.

4. Please define how EOP was calculated in the figure legends and/or methods section.

This has been added to the methods section and can be found in lines 833-834:

"EOP values were assessed qualitatively given that strong defense prevents formation of individual plaques. For one exception, Extended Data Fig. 7b, we used the top agar overlay method with different strains of interest and quantified plaques for three independent biological replicates."

5. For Fig. 4a, the 's' symbol within certain squares is not defined.

We apologize for this oversight; the "S" denotes smaller plaques and this is now labeled in Fig. 4a and Extended Data Fig. 6c, as well as the figure legends for these figures.

6. Figure 5 presentation could be improved to highlight the modularity of SNIPE. The use of the pie chart obscures/underwhelms many of the findings. Could representative examples of SNIPE with the unique domain architecture be shown using cartoon depictions? Or perhaps demonstrating the structural similarity between the N-term and some of the hits listed in the pie chart? Especially the non-TM group would be interesting.

We thank the reviewer for this great suggestion and have now updated Fig. 5d to include six representative examples of SNIPE homologs with different N-terminal architectures (three with transmembrane domains, and three without).

7. Within the section entitled “SNIPE interacts with the mannose permease complex” the line “To our knowledge, this is the first documented example of a phage protein interacting with a bacterial inner membrane protein during genome infection” is unclear and unsupported. I think the authors mean “...interacting with a an antiphage inner membrane protein...”. Also, further work is required to verify this SNIPE directly interacts with ManYZ.

We appreciate the reviewer’s comment and agree that our original statement was both unclear and potentially overreaching. To avoid confusion and ensure the manuscript remains well supported by the available evidence, we have removed this statement from the revised version of the manuscript.

8. Figure 3c is misleading. The authors are comparing protein abundance plus and minus phage infection, however, because they are measuring abundance of a phage protein and there is only phage protein in one sample, the graph artificially illustrates an enrichment of TMP. The more accurate comparison is in Fig. 3e and 3c should be removed or placed in the supplement.

We thank the reviewer for this important point. We have now moved what was Fig. 3c (TurboID-ManZ SNIPE-GFP +/- phage infection) to the supplement. Additionally, we have reorganized the logic of the main text to start with comparing TurboID-ManZ with FtsH-TurboID and ProW-TurboID to emphasize that TurboID-ManZ labeling of SNIPE-GFP and the TMP is specific. Then, we describe the additional experiments testing the dependence of this specific labeling on phage infection and SNIPE-GFP expression. This updated logic is found in lines 227-242 and is copied below.

"To perform proximity labeling during phage genome injection, we added wild-type λ and exogenous biotin to cells producing ManXY, TurboID-ManZ, and SNIPE-GFP. At 15 minutes post-infection, we lysed cells and used streptavidin pulldowns to isolate biotinylated proteins, which were identified by mass spectrometry. As expected, one of the most enriched proteins was AccB, the only protein that is naturally biotinylated in *E. coli* (Fig. 3c, Extended Data Fig. 4d). We also observed strong enrichment of ManX, a known interaction partner of ManYZ, providing a proof of concept for this assay; ManY was not enriched, but only harbors one cytoplasmic lysine that could be biotinylated²⁸. Notably, SNIPE-GFP was also enriched, and the only phage protein that was robustly labeled by TurboID-ManZ during genome injection was the λ tape measure protein (TMP), also termed gpH (additional tail components gpJ and gpZ were weakly labeled in some TurboID-ManZ samples). To assess the specificity of these interactions, we fused TurboID to one of two other inner membrane proteins, ProW or FtsH, and again performed proximity labeling during λ infection. We found that ManX, SNIPE-GFP, and

the λ tape measure protein were enriched only in the TurboID-ManZ samples (Fig. 3c, Extended Data Fig. 4c-e). These results suggest that ManYZ specifically associates with SNIPE and the λ tape measure protein during genome injection.

To test whether SNIPE interacts with ManYZ independent of the tape measure protein, we performed proximity labeling with TurboID-ManZ in the presence and absence of phage infection. Indeed, we observed strong enrichment of SNIPE-GFP both in the presence and absence of phage infection, and similar results were obtained in a similar experiment involving untagged SNIPE (Extended Data Fig. 4f-g). These observations support the notion that SNIPE and ManYZ interact *in vivo* prior to and during λ infection. Notably, this interaction did not impair ManYZ-mediated transport of mannose across the inner membrane, as SNIPE-containing cells were able to metabolize mannose on MacConkey agar (Extended Data Fig. 4h). We also asked whether the tape measure protein interacts with ManYZ independently of SNIPE. During genome injection, we observed that TurboID-ManZ labeled the tape measure protein at similar levels in the presence and absence of SNIPE (Extended Data Fig. 4i). Together, these data suggest that ManYZ interacts with both SNIPE and the tape measure protein, independent of each other. By extension, our results indicate that SNIPE interacts with ManYZ prior to infection, positioning it to also associate with the tape measure protein during infection and thereby target the incoming phage DNA for degradation."

9. Please provide western blot analysis of the SNIPE mutants used throughout (Δ DUF4041, Δ TM, etc.) to demonstrate that protein expression levels are not affected. These data are crucial to validate the authors conclusions.

We have now provided western blots for tagged SNIPE constructs, which can be found in Extended Data Fig. 2b, f, as well as Extended Data Fig. 4c.

Referees' comments:

Referee #1 (Remarks to the Author):

I appreciate that the authors have added new information. I have a few additional comments (see below).

1- Regarding the interaction of SNIPE with the mannose operon, why growth curves with mannose were not performed? I would have expected growth curves in minimal medium with mannose at the sole carbon source (and with glucose as a control). The MacConkey agar assay, indicate that the mannose operon is still functional but I was wondering of its efficacy might be impair.

We have performed the suggested experiment using minimal media with glucose or mannose, which can now be found in Extended Data Fig. 4i. The results are consistent with the MacConkey agar assay and further suggest that SNIPE does not impair transport of mannose. Extended Data Fig. 4h-i are provided below for reference.

2- MOI and spherical cells, at which MOIs did you observed this phenomenon? It is unclear why the plasmid transformation data is not added in supp mat?

We have added the plasmid transformation data to the supplemental material, and it can be found in Extended Data Fig. 1a. The experiment is now referenced in lines 114-117 of the main text, which is copied below:

"Additionally, growth curve assays showed that SNIPE, but not PD- λ -3, permitted cell survival in the presence of high phage concentrations (Fig. 1b). These results confirm that

SNIFE provides direct defense, enabling cells to ward off infection without notably compromising cell growth. Notably, SNIFE did not block plasmid DNA transformation, indicating that its activity is specific to phage infection (Extended Data Fig. 1a)."

3- The information that you attempted but failed to isolate λ escapers by plating for single plaques and also performing liquid evolution experiments should be in the manuscript as data not shown.

We have now updated the main text to mention our failure to isolate λ escapers, which can be found in lines 257-259.

"In principle, SNIFE could target phage DNA if it associates with LamB or ManYZ. After unsuccessful attempts to isolate λ escape mutants, we turned to a "generalist" mutant of λ that evolved to infect Δ lamB and Δ manYZ strains²⁶."

Referee #2 (Remarks to the Author):

The authors have satisfied all of my critiques and I congratulate them on their high-quality work.